# MULTI-MARGINAL FLOW MATCHING WITH ADVERSARIALLY LEARNT INTERPOLANTS

**Oskar Kviman***
KTH

**Kirill Tamogashev***
University of Edinburgh

**Nicola Branchini**
University of Edinburgh

**Víctor Elvira**
University of Edinburgh

**Jens Lagergren**
KTH

**Nikolay Malkin**
University of Edinburgh
CIFAR Fellow

 **mmacosha/adversarially-learned-interpolants**

## ABSTRACT

Learning the dynamics of a process given sampled observations at several time points is an important but difficult task in many scientific applications. When no ground-truth trajectories are available, but one has only snapshots of data taken at discrete time steps, the problem of modelling the dynamics, and thus inferring the underlying trajectories, can be solved by multi-marginal generalisations of flow matching algorithms. This paper introduces a novel flow matching method that overcomes the limitations of existing multi-marginal trajectory inference algorithms. Our proposed method, ALI-CFM, uses a GAN-inspired adversarial loss to fit neurally parameterised interpolant curves between source and target points such that the marginal distributions at intermediate time points are close to the observed distributions. The resulting interpolants are smooth trajectories that, as we show, are unique under mild assumptions. These interpolants are subsequently marginalised by a flow matching algorithm, yielding a trained vector field for the underlying dynamics. ALI-CFM outperforms existing baselines on spatial transcriptomics and cell tracking problems, while performing on par with them on single-cell trajectory prediction, which showcases its versatility and scalability.

## 1 INTRODUCTION

Modelling the time-dependent dynamics of a system given experimental observations is a central task in many scientific problems in biology (see Schiebinger et al. (2019); Bunne et al. (2023)), medicine (see Oeppen and Vaupel (2002); Hay et al. (2021)), and other areas. The problem involves a collection of data snapshots taken at various time steps that together provide an empirical account of some process. Examples of such processes include recordings of health measurements, evolution of a disease (Waddington, 1942; Hay et al., 2021), and time-series single-cell RNA sequencing data (scRNA-seq; Macosko et al. (2015); Klein et al. (2015)).

Formally, a (deterministic) system in $\mathbb{R}^n$ can be described by an ordinary differential equation (ODE) $dx_t = v_t(x_t)\,dt$, where $v_t$ is a time-dependent vector field. In a bimarginal case where samples are given from a pair of marginal distributions $q_0, q_1$, we aim to find $v_t$ such that the ODE's integration map from $t = 0$ to $t = 1$ pushes $q_0$ to $q_1$. In the multi-marginal case, the marginal distributions $p_t$ induced by the dynamics with initial conditions $p_0 = q_0$ should also satisfy intermediate

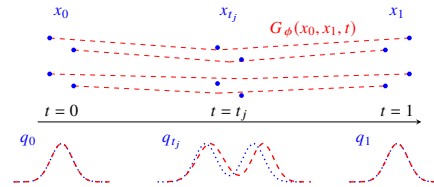

Figure 1: Adversarially learnt interpolants (red curves) follow pushforward distributions (red densities) that approximate the intermediate-time marginal distributions $q_{t_j}$ (blue) and, by construction, have the correct end-marginals $q_0$ and $q_1$.

---

*Equal contribution. Correspondence to k.tamogashev@sms.ed.ac.uk

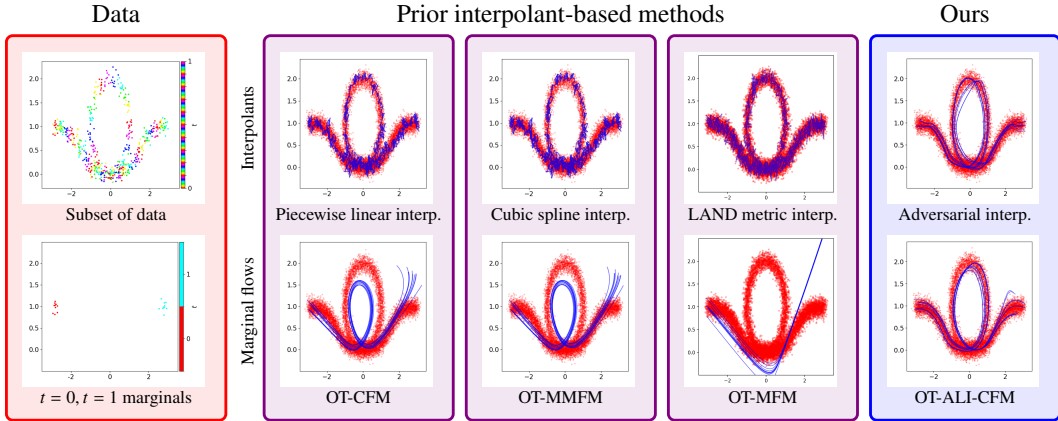

Figure 2: Comparison of CFM (Tong et al., 2024), MFM (Kapuśniak et al., 2024), MMFM (Rohbeck et al., 2025) and our ALI-CFM method on a synthetic 2D 'knot' distribution. See §4 for details.

conditions, namely, $p_{t_i} = q_{t_i}$ for a set of times $0 = t_1 < t_2 < \cdots < t_K = 1$. The intermediate distributions $q_{t_i}$ are provided by datasets of samples.

While this problem can be tackled by applying flow matching (Lipman et al., 2023; Albergo et al., 2025; Liu et al., 2023; Tong et al., 2024) to learn flows for every pair of consecutive marginals $q_{t_i}, q_{t_{i+1}}$ (Tong et al., 2024), we show that such approaches can lead to non-smooth interpolation curves. Recently, several methods specialising in multi-marginal problems have been proposed. Rohbeck et al. (2025) proposes to use cubic splines to build trajectories that pass through samples from intermediate distributions. However, spline interpolation methods do not seem to scale well to high dimensions (Lee et al., 2025). Another line of work (Neklyudov et al., 2024; Kapuśniak et al., 2024) assumes that the interpolants in each interval $[t_i, t_{i+1}]$ follow a certain geometry that can be learnt. These methods fit an interpolant and use it for learning the vector field of the underlying dynamics. However, the interpolants are still piecewise and are difficult to marginalise with precision, as we show in §4.1 and §4.2 (see, e.g., Fig. 2).

To overcome the limitations of previous methods, we propose to learn the interpolations between distributions using a GAN-like adversarial objective (Goodfellow et al., 2014; Huang et al., 2024) to obtain what we call adversarially learnt interpolants (ALIs). ALI directly matches the target intermediate marginals with those of the learnt interpolants (Fig. 1). The interpolants can then be marginalised by a conditional flow matching (CFM) method, allowing us to model complex time-dependent behaviour while using available intermediate-time information. The full algorithm, called ALI-CFM, approximates the data distribution at each time step, as opposed to prior methods that force interpolants to pass through samples explicitly, which makes our method especially useful in cases where the provided dataset is noisy.

Our core contributions in this paper are the following:

(1) We propose a novel technique for learning interpolants using a GAN-like adversarial objective. We show that the introduced algorithm can be easily used for multi-marginal problems with snapshots in either discrete or continuous time. We show the versatility of ALI-CFM on both synthetic and real-data examples.
(2) We successfully apply our method to a tumour coordinate inference problem using spatial transcriptomics (ST) data, where our method significantly outperforms all the existing baselines.
(3) We demonstrate the scalability of ALI-CFM on a scRNA-seq trajectory inference problem.

## 2 METHOD

**Background.** We summarise CFM, roughly following the setting and exposition of Lipman et al. (2023); Tong et al. (2024); Pooladian et al. (2023). All statements below hold under regularity conditions whose details are not relevant in this paper.

We assume a dynamical system in $\mathbb{R}^n$ given by an ordinary differential equation (ODE) $dx_t = v_t(x_t)\,dt$. Its integration map $\psi_t : \mathbb{R}^n \to \mathbb{R}^n$ from time 0 to time $t$ satisfies

$$\frac{d}{dt}\psi_t(x_0) = v_t(\psi_t(x_0)), \quad \psi_0(x_0) = x_0. \tag{1}$$

The integration map, together with stochastic initial conditions $x_0 \sim p_0$, defines a probability path $p_t = (\psi_t)_\# p_0$, where $p_t$ is the marginal distribution of $x_t$.

In the bimarginal FM setting, one observes samples from two marginal distributions, $x_0 \sim q_0$ and $x_1 \sim q_1$, and fixes a vector field $v_t$ such that its integration map $\psi_t$ satisfies $(\psi_1)_\# q_0 = q_1$. (This vector field is not tractably computable, but is described by interpolants, as we discuss below.) Having fixed $v_t$ as a target, the goal is to approximate it with a neural net $u_t^\theta$ with weights $\theta$. This could be done via the FM objective

$$L_{\text{FM}} = \mathbb{E}_{t \sim U[0,1], x \sim p_t} \| u_t^\theta(x) - v_t(x) \|_2^2. \tag{2}$$

However, since $v_t$ and $p_t$ are intractable to compute and sample from, the loss (2) is intractable.

Instead, one assumes a family of *interpolant* curves, one for every pair $x_0, x_1$ in the support of $q_0 \otimes q_1$. These curves are denoted $G(x_0, x_1, t)$ and should satisfy $G(x_0, x_1, t) = x_t$ for $t = 0, 1$. For example, a linear interpolant is described by $G(x_0, x_1, t) = tx_1 + (1 - t)x_0$. We define $v_t(x_t \mid x_0, x_1) := \frac{d}{dt}G(x_0, x_1, t)$. For any joint distribution $\pi$ over $\mathbb{R}^n \times \mathbb{R}^n$ whose marginals are $q_0$ and $q_1$, respectively, it can then be shown that the marginal vector field

$$v_t(x_t) := \mathbb{E}\left[v_t(x_t \mid x_0, x_1) \mid x_t = G(x_0, x_1, t)\right], \quad (x_0, x_1) \sim \pi, \tag{3}$$

pushes $q_0$ to $q_1$ and can thus be used as the learning target for $u_t^\theta$. The marginals $p_t$ of the resulting dynamics are tractably sampled by drawing $(x_0, x_1) \sim \pi$ and setting $x_t = G(x_0, x_1, t)$. While $v_t$ itself is still not tractable, one can replace (2) by the following *conditional flow matching* (CFM) objective:

$$L_{\text{CFM}} = \mathbb{E}_{t \sim U[0,1], (x_0, x_1) \sim \pi} \| u_t^\theta(x_t) - v_t(x_t | x_0, x_1) \|_2^2, \text{ where } x_t = G(x_0, x_1, t). \tag{4}$$

It is easy to show that the gradients of (4) and (2) coincide, and (4) thus provides a tractable way to learn the target vector field (Lipman et al., 2023).

The above setting leaves two choices open: the **coupling** $\pi$ and the **interpolants** $G$. Past work has proposed an independent coupling $\pi = q_0 \otimes q_1$, giving objectives equivalent to those in Liu et al. (2023); Albergo et al. (2025), or couplings computed by (possibly minibatch or entropic) optimal transport (Tong et al., 2024; Pooladian et al., 2023). The latter has been shown to result in straighter integration curves and solve the dynamic optimal transport problem. For the interpolants, linear and trigonometric (Albergo et al., 2025) curves have been proposed, as well as those trained to pass through areas of high data distribution density (Kapuśniak et al., 2024). In the multi-marginal setting, piecewise linear (Tong et al., 2024) and cubic spline (Rohbeck et al., 2025) interpolants have been used. See §3 for a discussion.

## 2.1 Adversarial learning of interpolants

We move from the bimarginal to the multi-marginal setting, where data has been collected from a sequence of $K$ marginal distributions, $x_{t_i} \sim q_{t_i}$ with corresponding time stamps $0 = t_1 < t_2 < \cdots < t_K = 1$. Given a coupling $\pi$ between $q_0$ and $q_1$ and a neural network $f_\phi$, we consider neural interpolants (Neklyudov et al., 2024; Kapuśniak et al., 2024)

$$G_\phi(x_0, x_1, t) = (1 - t)x_0 + tx_1 + t(1 - t)f_\phi(x_0, x_1, t). \tag{5}$$

Our aim is to match the intermediate distributions at $t_i$ of the interpolants when $(x_0, x_1) \sim \pi$ to the given marginals $q_{t_i}$, that is, to enforce

$$(G_\phi(\cdot, \cdot, t_i))_\# \pi = q_{t_i}. \tag{6}$$

(The parametrisation (5) guarantees $G_\phi(x_0, x_1, t) = x_t$ for $t = 0, 1$, so (6) holds automatically for $i \in \{1, K\}$.) In order to approximately enforce (6), we use an adversarial learning scheme. Let $D_\gamma(x_t, t)$ be a second neural network, tasked to discriminate between marginal samples, $x_t \sim q_t$, and the learnable interpolants in (5). Optimising the min-max GAN objective (Goodfellow et al., 2014) for each $t_i$,

$$\min_{G_\phi} \max_{D_\gamma} \underbrace{\mathbb{E}_{(x_0, x_1) \sim \pi}\left[\log(1 - D_\gamma(G_\phi(x_0, x_1, t_i), t_i))\right] + \mathbb{E}_{q_{t_i}}\left[\log D_\gamma(x_{t_i}, t_i)\right]}_{L_{\text{GAN}}(G_\phi, D_\gamma; t_i)}, \tag{7}$$

is then equivalent, under the assumption of an optimal discriminator, to minimising the Jensen-Shannon divergence (Goodfellow et al., 2014) between $q_{t_i}$ and, in this case, $G_\phi(\cdot, \cdot, t_i)_{\#}\pi$. Notably, our 'generators' – the interpolants – are conditioned on scalar-valued time inputs, associated with the targeted $q_{t_i}$. While the noise in GANs typically comes from a fixed distribution, in (7) the pair $(x_0, x_1) \sim \pi$ plays the role of 'noise'.

The solutions to the min-max problem in (7) are not unique and can induce arbitrarily curved interpolants. To this end, we propose regularising terms, $L_{\text{reg}}(G_\phi; t)$, in the learning objective that guarantee unique interpolants. See §2.3 for explicit formulations of the proposed regularisers. In Algorithm 1 we provide pseudocode for training ALIs.

---

**Algorithm 1** Training adversarially learnt interpolants

---

**Require:** coupling $\pi$, trainable correction network $f_\phi$, trainable discriminator $D_\gamma$, regulariser $L_{\text{reg}}$, regularising weight $\lambda$
1: **while** Training **do**
2:     Draw $(x_0, x_1) \sim \pi$, $i \sim \text{Unif}(\{2, ..., K-1\})$, $x_{t_i} \sim q_{t_i}$
3:     $G_\phi(x_0, x_1, t_i) \leftarrow (1 - t_i) x_0 + t_i x_1 + t_i(1 - t_i) f_\phi(x_0, x_1, t_i)$        ▷ Equation (5)
4:     $\widehat{L}_{\text{GAN}} \leftarrow$ estimate GAN loss with $x_{t_i}$ and $G_\phi(x_0, x_1, t_i)$        ▷ Equation (7)
5:     $\widehat{L}_{\text{reg}} \leftarrow$ estimate regularising term        ▷ Regularisers are suggested in §2.3
6:     $\widehat{L}_{\text{ALI}} \leftarrow \widehat{L}_{\text{GAN}} + \lambda \widehat{L}_{\text{reg}}$        ▷ Equation (8)
7:     Update $\phi$ and $\gamma$ using gradient $\nabla_{\phi,\gamma} \widehat{L}_{\text{ALI}}$
8: **return** adversarially learnt interpolants $G_\phi$

---

## 2.2 COMPLETE METHOD: ALI-CFM

The loss (7) and the above result motivate the full *adversarially learnt interpolants (ALI) objective*:

$$L_{\text{ALI}}(G_\phi, D_\gamma) = \mathbb{E}_{i \sim \text{Unif}(\{2,...,K-1\})} \left[ L_{\text{GAN}}(G_\phi, D_\gamma; t_i) + \lambda L_{\text{reg}}(G_\phi; t_i) \right], \quad (8)$$

where $\lambda > 0$ is a regularisation weight.

Once the interpolants $G_\phi$ have been trained using ALI, they can be marginalised using the CFM objective (4), using the same coupling $\pi$.[1] At convergence, this yields a dynamical system, defined by a vector field $u_t^\theta$, whose marginals $p_t$ at each $t$ match the interpolants' marginals $(G_\phi(\cdot, \cdot, t))_{\#}\pi$. If further (6) has been satisfied, then the resulting flow solves the multi-marginal transport problem: $p_{t_i} = q_{t_i}$ for all $i$.

We refer to the complete method as ALI-CFM: it consists of (i) learning interpolants $G_\phi$ using ALI (Algorithm 1), then (ii) marginalising them to yield the time-dependent vector field $u_t^\theta$ (Algorithm 2). The prefixes I- and OT- in the algorithm names specify if the coupling $\pi$ used is independent or a (minibatch) optimal transport plan, respectively.

---

**Algorithm 2** Training ALI-CFM

---

**Require:** coupling $\pi$, trained ALI $G_\phi$, trainable CFM net $u_t^\theta$
1: **while** Training **do**
2:     $(x_0, x_1) \sim \pi$, $t \sim U[0, 1]$
3:     $G_\phi(x_0, x_1, t) \leftarrow (1 - t) x_0 + t x_1 + t(1 - t) f_\phi(x_0, x_1, t)$        ▷ Equation (5)
4:     $\frac{d}{dt} G_\phi(x_0, x_1, t) \leftarrow x_1 - x_0 + t(1 - t) \frac{d}{dt} f_\phi(x_0, x_1, t) + (1 - 2t) f_\phi(x_0, x_1, t)$
5:     $\widehat{L}_{\text{ALI-CFM}} \leftarrow \|u_t^\theta(G_\phi(x_0, x_1, t)) - \frac{d}{dt} G_\phi(x_0, x_1, t)\|^2$        ▷ Estimate CFM loss
6:     Update $\theta$ using gradient $\nabla_\theta \widehat{L}_{\text{ALI-CFM}}$
7: **return** learned vector field $u_t^\theta$

---

[1]The objective requires differentiating the learnt interpolants with respect to $t$, which is done with little overhead using autograd. Note that the interpolants' parameters $\phi$ are fixed at this stage.

## 2.3 REGULARISERS FOR INTERPOLANTS

In this section we propose three regularising terms to use in (8) with provable uniqueness guarantees for the first two options. See §4.3 for an ablation study of the different regularisers.

**Linear reference regulariser.** Here we introduce a regularising term in the ALI learning objective which penalises deviations from the linear interpolant between coupled samples from the end-marginals. Letting $\ell(x_0, x_1, t) = (1 - t)x_0 + tx_1$, we define

$$L_{\text{reg}}(G_\phi; t_i) = \mathbb{E}_{(x_0, x_1) \sim \pi} \left[ \|G_\phi(x_0, x_1, t_i) - \ell(x_0, x_1, t)\|^2 \right]. \tag{9}$$

The problem of matching marginals (6) while minimising the regulariser (9) enjoys unique solutions:

**Theorem 2.1.** *Fix $t \in (0, 1)$, $q_t$, and a coupling $\pi$ between $q_0$ and $q_1$ such that the distribution $\ell(\cdot, \cdot, t)_{\#}\pi$ is absolutely continuous (a.c.) w.r.t. the Lebesgue measure. Then the interpolant $G(\cdot, \cdot, t) : \mathbb{R}^d \times \mathbb{R}^d \to \mathbb{R}^d$ minimising*

$$\mathbb{E}_{(x_0, x_1) \sim \pi} \|G(x_0, x_1, t) - \ell(x_0, x_1, t)\|^2 \tag{10}$$

*subject to $G(\cdot, \cdot, t)_{\#}\pi = q_t$, exists and is unique on the support of $\pi$ up to almost-everywhere equality.*

We provide the proof in Appendix A. Note that the assumption of an a.c. target interpolant is satisfied under a number of conditions, e.g., if $\pi$ is the solution to entropic OT with squared-euclidean cost (or either of its limiting cases: (nonentropic) OT or the independent coupling) and either $q_0$ or $q_1$ is a.c.

**Piecewise linear reference regulariser.** In settings where the supports of the intermediate marginal distributions differ from those of the end marginals, the linear reference from the previous section may restrict $G_\phi$ from accurate modelling of the target distributions, especially in high dimensions. Also, when $K$ is small, the learning of $f_\phi$ may benefit from objectives evaluated on the pseudo-time unit interval, as opposed to on a discrete set of time stamps. As such, we propose a second regulariser that regresses $G_\phi$ to a piecewise linear interpolant, marginalised over $t \in [0, 1]$

$$L_{\text{reg}}(\phi; t_i) = \mathbb{E}_{t \sim U[0,1]} \mathbb{E}_{(x_1, x_{t_i}, x_0) \sim \pi_{t_i}} \left[ \|G_\phi(x_0, x_1, t) - \ell(x_t | x_0, x_1, x_{t_i}, t)\|^2 \right], \tag{11}$$

where we let $\pi_{t_i}$ be a Markov-chained OT coupling (Tong et al., 2024) $\pi_{t_i} = \pi(x_1 | x_{t_i}) \pi(x_{t_i} | x_0) q_0(x_0)$, with

$$\ell(x_t | x_0, x_1, x_{t_i}, t) = \begin{cases} \frac{t x_{t_i} + (t_k - t) x_0}{t_i}, & t \leq t_i, \\ \frac{t x_1 + (1 - t) x_{t_i}}{1 - t_i}, & t > t_i, \end{cases} \tag{12}$$

and which comes with the following uniqueness guarantee.

**Theorem 2.2.** *Fix $t \in (0, 1)$, $q_t$, and a Markov–chained OT coupling $\pi_{t_i} = \pi(x_1 \mid x_{t_i}) \pi(x_{t_i} \mid x_0) q_0(x_0)$ between $q_0$ and $q_1$. Let $\ell(x_t \mid x_0, x_1, x_{t_i}, t)$ be the piecewise linear reference map in (12), and assume that the distribution $\ell(\cdot \mid \cdot, \cdot, \cdot, t)_{\#}\pi_{t_i}$ is a.c. with respect to the Lebesgue measure. Then the interpolant $G(\cdot, \cdot, t) : \mathbb{R}^d \times \mathbb{R}^d \to \mathbb{R}^d$ minimising*

$$\mathbb{E}_{(x_0, x_{t_i}, x_1) \sim \pi_{t_i}} \|G(x_0, x_1, t) - \ell(x_t \mid x_0, x_1, x_{t_i}, t)\|^2 \tag{13}$$

*subject to $G(\cdot, \cdot, t)_{\#}\pi = q_t$, where $\pi$ is the $(x_0, x_1)$-marginal of $\pi_{t_i}$, exists and is unique on the support of $\pi$ up to almost-everywhere equality. Moreover, if $t$ is averaged according to $U[0, 1]$ as in (11),*

$$L_{\text{reg}}(G; t_i) = \mathbb{E}_{t \sim U[0,1]} \mathbb{E}_{(x_0, x_{t_i}, x_1) \sim \pi_{t_i}} \|G(x_0, x_1, t) - \ell(x_t \mid x_0, x_1, x_{t_i}, t)\|^2, \tag{14}$$

*the same $G(\cdot, \cdot, t)$ minimises $L_{\text{reg}}$ for $U$-a.e. $t$.*

See Appendix A for a proof.

**Norm of the second derivative as a regulariser.** An alternative way to introduce regularisation is to add a component that would enforce a smoothness constraint on the entire interpolant. One way to do that is to use the integral of the norm of the second derivative of the interpolant. A similar idea motivated the use of cubic splines in Rohbeck et al. (2025). In contrast, we directly incorporate the integral of the norm of the second derivative as a regularisation term:

$$L_{\text{reg}}(\phi) = \mathbb{E}_{(x_0, x_1) \sim \pi} \left[ \int_0^1 \left\| \frac{\partial^2}{\partial t^2} G_\phi(x_0, x_1, t) \right\|_2^2 dt. \right] \tag{15}$$

Computing the second derivative of the neural network exactly would be prohibitively costly. Instead, we use a numerical approximation of the second derivative:

$$\frac{\partial^2}{\partial t^2} G_\phi(x_0, x_1, t) \approx \frac{G_\phi(x_0, x_1, t + h) + G_\phi(x_0, x_1, t - h) - 2 \cdot G_\phi(x_0, x_1, t)}{h^2}. \tag{16}$$

We found a Monte Carlo estimate of the integral using three samples of $t \sim U[0, 1]$ to be sufficient.

## 3 RELATED WORKS

Here, we describe previously proposed multi-marginal flow matching methods and relate them to our novel ALIs. We begin by stressing a distinction between our approach and all existing approaches: ALIs are interpolants that follow distributions which match the intermediate-time marginals, $q_{t_i}$, in distribution (see Fig. 1), while all existing multi-marginal interpolation methods match the intermediate marginals pointwise.

The work of Kapuśniak et al. (2024); Neklyudov et al. (2024) is closest to our method, as they also learn nonlinear interpolants parameterised by a time-dependent neural network. However, there are differences in both the algorithms and the underlying motivations. Notably, neither of these approaches explicitly approximates a divergence between the intermediate time marginals and the interpolants, nor do they use adversarial training. Kapuśniak et al. (2024) motivate their approach, metric flow matching (MFM), by addressing distributions that are supported on a manifold with a metric learning approach (Cox and Cox, 2008; Xing et al., 2002). The metric used in MFM can be chosen as either time-independent or time-dependent. The former setting trivially contrasts to ALIs as our generator and discriminator networks are time-dependent, while the latter case results in piecewise interpolants (the interpolants are conditioned to pass through the marginal samples), which again differs from our distribution-fitting approach. Furthermore, our method does not rely on specifying a particular underlying metric, and, importantly, we show experimentally in §4.1 and §4.2 how time-independent metrics prevent MFM from learning valuable geodesics when the geometry of the underlying dynamics changes with time.

Rohbeck et al. (2025); Lee et al. (2025) propose an explicit parametrisation of interpolants using splines. However, these methods – along with the ones discussed before (Tong et al., 2024; Neklyudov et al., 2024; Kapuśniak et al., 2024) – assume that the obtained interpolants should pass through the available samples of marginal distributions. In addition to that, spline-based interpolants (Rohbeck et al., 2025; Lee et al., 2025) may scale poorly in high dimensions.

## 4 EXPERIMENTS

We conduct four different experiments. To showcase the flexibility of our ALIs to solve multi-marginal problems for large $K$, with noisy data, and where the data geometry changes with time, we consider a toy experiment in §4.1 and an experiment with real cell tracking data in §4.2. In §4.3 and §4.4, we experiment with scRNA-seq and ST data, respectively, where $K \in [4, 5]$. In all experiments, we compare our method with the baselines in terms of earth mover's distances (EMDs) and/or visual verifications. Finally, note that all methods here are trained with the OT coupling (see §2), but that a comparison of I-ALI-CFM and OT-ALI-CFM is provided in Appendix D.2.

### 4.1 SYNTHETIC DATA

First, we showcase the flexibility of our ALI-CFM method on synthetic data: a sequence of 1,200 marginal distributions centred along a knot (Fig. 2). We obtain 10 samples from each marginal distribution. The experiment shows that ALI-CFM is the only method capable of accurately capturing the time-dependent geometry of the 'knot' distribution (formalised in Appendix B). For OT-MFM, we considered the time-dependent LAND metric, where a sequence of LAND metrics is constructed from the data at $t_i$ and $t_{i+1}$, and the interpolants are learnt using all the data. More experimental details are provided in Appendix D.1.

Notably, the piecewise linear interpolant (Tong et al., 2024), the cubic spline interpolants (Rohbeck et al., 2025) and OT-MFM with time-dependent LAND metrics are non-smooth, which results in the

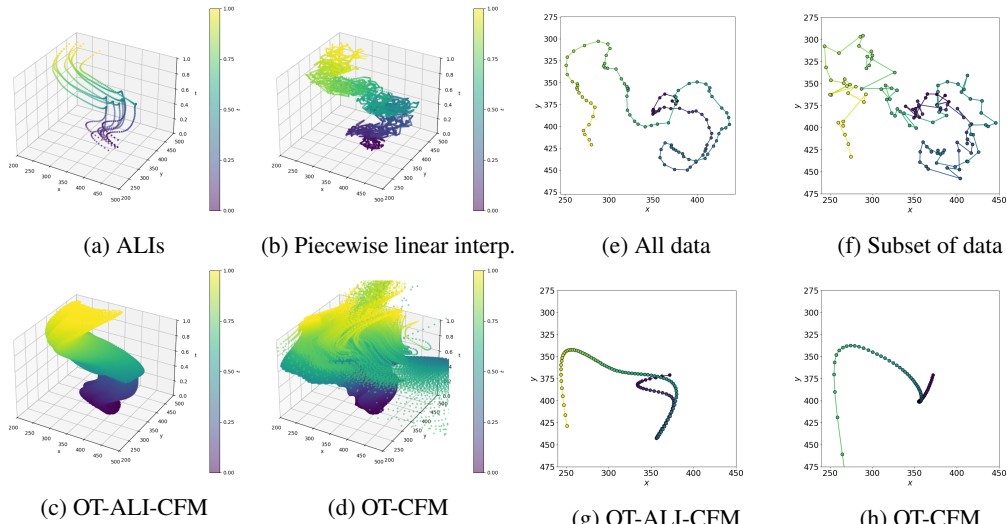

Figure 3: **(a)** and **(b)** show ten ALIs and piecewise linear interpolants, respectively, based on the subsampled cell tracking data, while **(c)** and **(d)** depict the resulting OT-ALI-CFM and OT-CFM vector fields. In the 2D figures we plot the centroids (one per frame) of **(e)** all of the segmentation data, **(f)** the subset training data, **(g)** the OT-ALI-CFM trajectories and **(h)** the OT-CFM trajectories. Note that the OT-CFM trajectories diverge in **(d)** and **(h)**.

corresponding CFM objective having high gradient variance. This was not the case when training OT-ALI-CFM on the smooth ALI interpolants. In Fig. 8 we visually compare the smoothness of ALIs and OT-MFM interpolants for different choices of $K$, demonstrating that MFM scales poorly with $K$, while our ALIs consistently produce smooth and accurate interpolants.

## 4.2 LEARNING SPATIO-TEMPORAL CELLULAR DYNAMICS WITH CELL TRACKING DATA

Encouraged by our results in Fig. 2, we aim to learn the spatio-temporal dynamics of the positions of a cell. That is, given segmentation masks of a glioblastoma-astrocytoma U373 cell collected in 115 times steps/consecutive frames, provided by The Cell Tracking Challenge (Maška et al., 2023), we consider learning vector fields that, when integrated, produce trajectories that follow the tracked cell movements. This is challenging as the shape of the cell contracts and expands while it moves along a loopy path (Fig. 3e).

To demonstrate ALIs' capacity to handle noisy datasets, we independently subsample ten segmentation coordinates per time stamp. Based on the subset, we learn both interpolants and CFMs. Then, we use all $x_0$ samples when integrating the vector field, i.e., we push all $x_0$ samples in the dataset through the learnt vector fields with 115 integration steps (same as the number of frames). Given the similar performances of OT-CFM and OT-MMFM in Fig. 2, we compare OT-ALI-CFM to OT-CFM and OT-MFM in this experiment. Note that, here, we implemented two versions of OT-MFM using the time-independent and time-dependent LAND metric. The latter setting is the same as in §4.1, while the former mimics the LiDAR experiment setup in Kapuśniak et al. (2024), where the metric is inferred from all the data, while the interpolants are learnt using only samples from $q_0$ and $q_1$.

Inspecting the resulting interpolants in Fig. 3, ALIs provide a smooth map between $(x_0, x_1)$ pairs while the piecewise linear interpolants, again, introduce instability reflected as non-smoothness in the marginalised trajectories. Therefore, we could not train an OT-CFM without the integrated vector fields diverging, whereas fitting an OT-ALI-CFM was straightforward. In Fig. 4, we overlay the training data and the inferred CFM trajectories on the microscopy images, visualising also the inability of time-independent OT-MFM to learn accurate dynamics when the data geometry changes with time. The plots in Fig. 10 support the latter observation. Meanwhile, the time-dependent OT-MFM produces similar interpolants to OT-CFM, which are not smooth due to their piecewise nature, causing the vector field integration to diverge (Fig. 11). We constrained the analysis here to visual evaluations as the results in the mentioned figures are conclusive.

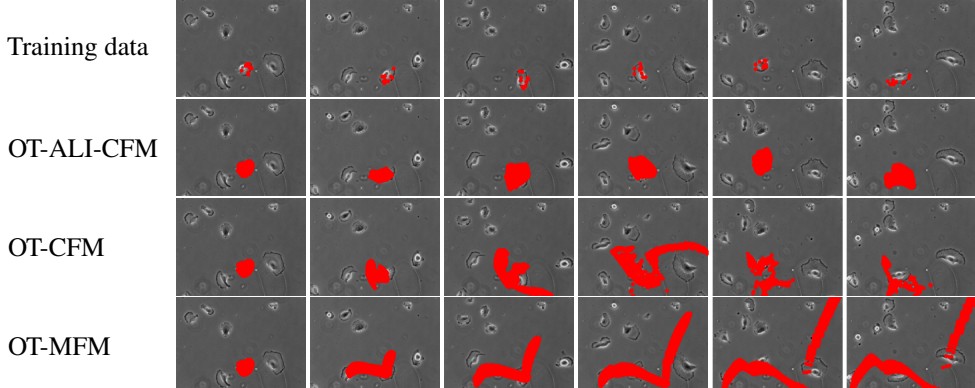

Training data

OT-ALI-CFM

OT-CFM

OT-MFM

Figure 4: From left to right we visualize frame 1, 25, 50, 75, 100 and 115 in the sequence of microscopy images. From top to bottom we overlay the images with the subsampled training data (10 samples per frame), and all segmentation samples $x_0$ pushed to the frame-specific times via the marginalised vector fields of OT-ALI-CFM, OT-CFM and OT-MFM. In concordance with the centroid plot in Fig. 3h, the trajectories from OT-CFM eventually diverge and become inaccurate. Meanwhile the time-independent metric in OT-MFM results in trajectories that do not capture the cell positions' temporal dependence.

Table 1: Trajectory inference on 5D PCA scRNA-seq data. Accuracy measured in EMD (smaller is better) w.r.t. the left-out marginal distributions, averaged over five independent runs.

| Algorithm ↓ Dataset → | Cite | EB | Multi |
|---|---|---|---|
| I-CFM | $1.236_{\pm0.050}$ | $1.156_{\pm0.42}$ | $1.150_{\pm0.091}$ |
| OT-CFM | $1.142_{\pm0.085}$ | $0.809_{\pm0.16}$ | $0.975_{\pm0.045}$ |
| OT-MFM | $0.793_{\pm0.019}$ | $0.711_{\pm0.050}$ | $0.890_{\pm0.123}$ |
| I-MMFM (Cubic splines) | $2.068_{\pm0.390}$ | $4.740_{\pm0.650}$ | $1.528_{\pm0.040}$ |
| OT-MMFM (Cubic splines) | $1.099_{\pm0.043}$ | $3.530_{\pm0.194}$ | $1.807_{\pm0.085}$ |
| OT-ALI-CFM (ours) | $0.910_{\pm0.024}$ | $0.742_{\pm0.022}$ | $0.925_{\pm0.018}$ |

To produce Fig. 3, we trained the CFM nets for 50,000 iterations with a learning rate of $10^{-3}$ and minibatches of size 128. The CFM nets were 3-hidden-layer MLPs with 256 units in each layer and SELU activations. ALI was regularised with $\lambda = 1$ and (9). More details on data and hyperparameters are provided in D.2.

## 4.3 SINGLE-CELL TRAJECTORY INFERENCE

As another real-world example, we experiment with time-series single-cell data, predicting the gene expression levels of cell populations that evolve over time. The samples from the $K$ consecutive populations are naturally unpaired, and the dynamics of the gene expressions are unknown. We follow preceding works (Tong et al., 2024) on predicting the marginal distributions of held-out intermediate-time data, measuring accuracy via the earth-mover distance (EMD) between the CFM trajectories and the held-out data at the corresponding time stamp. That is, using the remaining data, we train ALIs by optimising the objective in (8) and then learn ALI-CFMs by minimising (4). In Tables 1 and 2, we compare the performance of OT-ALI-CFM to other CFM methods on Embryoid body (EB; Moon et al. (2019)) data, and on Cite-seq (Cite) and Multiome (Multi) data from Lance et al. (2021). The preprocessing pipeline of single-cell data closely follows that of Kapuśniak et al. (2024). The only difference is that we normalise the data to ensure stability during training and denormalise only for metric computations. More details are given in Appendix D.3, where we also share an ablation study of the effect of our regularisers on the trajectory curvature and on the EMDs.

Although our algorithm is on par with existing baselines, we believe that the nature of adversarial training makes it difficult to completely outperform OT-MFM. Since our adversarially learnt interpolant matches the points at each time step in a distribution-matching sense, it might lose in pointwise metrics to methods that are trained to overfit to the given data.

Table 2: Trajectory inference on 50D and 100D PCA scRNA-seq data. The accuracies here are measured in the same way as in Table 1.

| Dim. → | 50 | | 100 | |
|---|---|---|---|---|
| Algorithm ↓ — Dataset → | Cite | Multi | Cite | Multi |
| I-CFM | $42.478_{\pm 0.930}$ | $51.098_{\pm 0.340}$ | $49.929_{\pm 0.391}$ | $57.801_{\pm 0.365}$ |
| OT-CFM | $38.367_{\pm 0.295}$ | $47.205_{\pm 0.184}$ | $45.148_{\pm 0.207}$ | $54.630_{\pm 0.456}$ |
| I-MFM | $41.172_{\pm 0.269}$ | $48.415_{\pm 0.793}$ | $46.339_{\pm 0.618}$ | $53.667_{\pm 0.768}$ |
| OT-MFM | $36.471_{\pm 0.480}$ | $45.879_{\pm 0.438}$ | $42.232_{\pm 0.249}$ | $51.169_{\pm 0.523}$ |
| OT-ALI-CFM (ours) | $41.449_{\pm 0.942}$ | $46.454_{\pm 0.538}$ | $48.496_{\pm 0.814}$ | $54.554_{\pm 0.832}$ |

## 4.4 TUMOUR COORDINATE INFERENCE WITH SPATIAL TRANSCRIPTOMICS DATA

Understanding cancer evolution is a key problem to solve for accurate tumor analysis and effective treatment development (Greaves and Maley, 2012). ST (Ståhl et al., 2016) has recently enabled studies of spatial tumor dynamics (Shafighi et al., 2024). In an ST platform, a thin section of a tissue is placed on a slide with coarsely distributed features, where cells in the tissue are sequenced, while maintaining the spatial locations of the cells. As such, given a single tissue section, any ST platform returns a 2D distribution of gene expression data, often accompanied by an hematoxylin and eosin (H&E) stained image of the section. Multiple sections are typically processed, allowing for informative analyses of the spatial organisation of cancer cells.

Table 3: Average EMD on ST data using learnt vector fields.

| Method | EMD (↓) |
|---|---|
| OT-CFM | $109.76_{\pm 9.98}$ |
| OT-MMFM | $109.17_{\pm 9.82}$ |
| OT-MFM | $183.88_{\pm 53.92}$ |
| OT-ALI-CFM | $\mathbf{98.91_{\pm 2.03}}$ |

We consider the breast cancer dataset published in Mo et al. (2024), using $K = 4$ sections, each processed with the Visium 10x Genomics platform. There are more than 1,000 samples in each section, each $x_t \in \mathbb{R}^2$, and the number of samples vary with $t$. We seek to model the distribution of tumour regions in an unobserved tissue section. As far as we are aware, this task has not previously been attempted, however it is well-suited to being tackled using FM as the samples (coordinates) in each section are naturally unpaired.

The four sections are spatially ordered along the $z$-axis, and the authors of the dataset have assigned the coordinates in each section with a tumour purity score based on the measured gene expressions. We classified all coordinates with a score above 0.8 as belonging to a tumour region[2] and aligned the sections (details in Appendix C), which enabled us to consider a common coordinate system across sections. We visualise the preprocessed data in Fig. 5 and the raw images in Fig. 7.

Following the notational convention, we let pseudo-times $t \in [0, 1]$ correspond to normalised $z$-coordinates, such that $q_0$ is the distribution of coordinate samples in the bottom section, and so on. We assume equal spacing between the sections, which is a reasonable assumption based on the documentation in Mo et al. (2024). We experiment with leave-one-out interpolation, omitting either section two ($t = 1/3$) or three ($t = 2/3$) from the training data. We then compare the EMD between the held-out data and the integrated vector fields. The EMD results are shown in Table 3, and a visualisation is provided in Fig. 6.

This dataset is challenging as the marginal distributions are highly multimodal, with modes that are not present in all sections. Also, tissues can shrink when stored before processing which makes alignment challenging (Appendix C; Zeira et al. (2022)). These challenges explain the OT-CFM and OT-MMFM results, while we believe that this makes the time-dependent LAND metric in OT-MFM inaccurate.

Interestingly, although GAN optimisation is considered sensitive to multimodal data distributions (Arjovsky et al., 2017; Huang et al., 2024), we found that our regulariser in (11) along with the Markov-chained OT coupling made training stable, with clearly impressive results.

All CFMs were trained using MLPs with three hidden layers, 128 units per layer and SELU activation functions. The CFMs were trained for 10,000 epochs with a batch size of 128. $D_\gamma$ and $G_\phi$ were

---

[2]The threshold was chosen based on visual agreement between the tumour-classified coordinates and the tumour regions in the corresponding H&E-stained images (darker regions typically imply higher density of tumour cells).

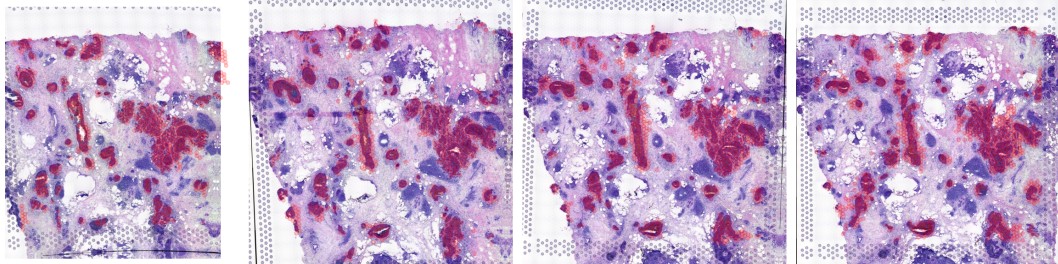

Figure 5: The aligned breast cancer H&E-stained images with overlaid scatter plots of tumor annotated coordinates (in light red).

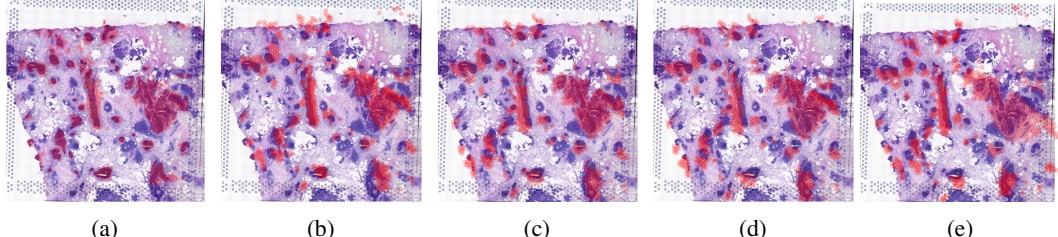

|  (a)  |  (b)  |  (c)  |  (d)  |  (e)  |

Figure 6: H&E-stained histology image of the third breast cancer tissue section overlaid with **(a)** true tumour annotated coordinates, and CFM trajectories integrated to $t = 2/3$ using **(b)** OT-ALI-CFM, **(c)** OT-CFM, **(d)** OT-MMFM and **(e)** OT-MFM. OT-ALI-CFM achieves the lowest EMD score w.r.t. to held-out coordinates in **(a)** (Table 3).

small MLP nets with two hidden layers, 64 units per layer and ELU activations. In the ALI objective (8) we used the piecewise-linear interpolant regulariser in (11). See Appendix D.4 for more details.

## 5    DISCUSSION

We propose ALI-CFM, a novel approach to learning interpolants in multi-marginal flow matching. ALI is a conceptually novel way to learn interpolations by matching the distributions of intermediate marginals, as opposed to the strict pointwise assignment used in all existing methods. We have demonstrated the superior capabilities of our method to fit complex dynamics with noisy samples from hundreds (§4.2) or thousands (§4.1) of marginal distributions – no other FM algorithm managed to demonstrate a comparable performance on this task. Moreover, across datasets and dimensions, our ALI-CFM performs on par with the state-of-the-art methods on single-cell trajectory inference.

Our interpolants are learnt using an adversarial learning objective, which fits well to the density-free problem setting in flow matching. Although GANs are challenging to train, we have proposed regularising terms to the objective that, apart from guaranteeing unique interpolants, empirically stabilise training.

In fact, compared to baselines (see the final paragraph in Appendix D.1), our adversarial training can reduce the complexity of training accurate interpolants in flow matching. Additionally, there is a rich literature on GAN optimisation, e.g., regarding alternative objectives (Arjovsky et al., 2017; Jolicoeur-Martineau, 2019; Sun et al., 2020; Huang et al., 2024).

## ACKNOWLEDGEMENTS

The authors are grateful to Alex Tong for helpful discussions about single-cell data preprocessing and methods from prior work. The authors also thank Hosein Toosi for providing valuable guidance regarding the preprocessing of the ST data.

The work of VE is supported by the Advanced Research and Invention Agency (ARIA). JL acknowledges support from the Swedish Research Council (project ID 2022-03516_VR). NM acknowledges support from the CIFAR Learning in Machines and Brains programme.

This work was enabled by the computational resources of the Edinburgh International Data Facility (EIDF) with funding from the Edinburgh Generative AI Laboratory (GAIL).

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

## A    PROOF OF UNIQUENESS THEOREMS

Here we restate the theorems from §2.3 and provide their proofs.

**Theorem 2.1.** *Fix $t \in (0, 1)$, $q_t$, and a coupling $\pi$ between $q_0$ and $q_1$ such that the distribution $\ell(\cdot, \cdot, t)_{\#}\pi$ is absolutely continuous (a.c.) w.r.t. the Lebesgue measure. Then the interpolant $G(\cdot, \cdot, t) : \mathbb{R}^d \times \mathbb{R}^d \to \mathbb{R}^d$ minimising*

$$\mathbb{E}_{(x_0, x_1) \sim \pi} \|G(x_0, x_1, t) - \ell(x_0, x_1, t)\|^2 \tag{10}$$

*subject to $G(\cdot, \cdot, t)_{\#}\pi = q_t$, exists and is unique on the support of $\pi$ up to almost-everywhere equality.*

*Proof of Theorem 2.1.* Let $G(\cdot, \cdot, t)$ be any function satisfying the constraint. Consider the following joint distribution on $\mathcal{X} \times \mathcal{X}$,

$$\Pi(dx_t, dx_t') = (\ell(\cdot, \cdot, t), G(\cdot, \cdot, t))_{\#}\pi. \tag{17}$$

The marginals of $\Pi$ over the first and second components are the a.c. distributions $\ell(\cdot, \cdot, t)_{\#}\pi$ and $q_t$, respectively. The distribution $\Pi$ is a Kantorovich plan between these two marginals whose cost is given by (10). This cost has a unique minimiser over all transport maps $\Pi$ (up to a.e. equality on the supports) because the first marginal is a.c., and the minimiser has $\Pi$ deterministic over the second component given the first, i.e., $\Pi$ is given by $(\text{Id}, T)_{\#}\ell(\cdot, \cdot, t)$ for some function $T : \mathcal{X} \to \mathcal{X}$.

This minimum is indeed uniquely achieved by $G = T \circ \ell - \ell$. Conversely, the optimal transport plan from $\ell(\cdot, \cdot, t)_{\#}\pi$ to $q_t$ yields an interpolant $G$ in the obvious manner, showing existence. $\square$

**Theorem 2.2.** *Fix $t \in (0, 1)$, $q_t$, and a Markov–chained OT coupling $\pi_{t_i} = \pi(x_1 \mid x_{t_i})\pi(x_{t_i} \mid x_0) q_0(x_0)$ between $q_0$ and $q_1$. Let $\ell(x_t \mid x_0, x_1, x_{t_i}, t)$ be the piecewise linear reference map in (12), and assume that the distribution $\ell(\cdot \mid \cdot, \cdot, \cdot, t)_{\#}\pi_{t_i}$ is a.c. with respect to the Lebesgue measure. Then the interpolant $G(\cdot, \cdot, t) : \mathbb{R}^d \times \mathbb{R}^d \to \mathbb{R}^d$ minimising*

$$\mathbb{E}_{(x_0, x_{t_i}, x_1) \sim \pi_{t_i}} \|G(x_0, x_1, t) - \ell(x_t \mid x_0, x_1, x_{t_i}, t)\|^2 \tag{13}$$

*subject to $G(\cdot, \cdot, t)_{\#}\pi = q_t$, where $\pi$ is the $(x_0, x_1)$-marginal of $\pi_{t_i}$, exists and is unique on the support of $\pi$ up to almost-everywhere equality. Moreover, if $t$ is averaged according to $U[0, 1]$ as in (11),*

$$L_{\text{reg}}(G; t_i) = \mathbb{E}_{t \sim U[0,1]} \mathbb{E}_{(x_0, x_{t_i}, x_1) \sim \pi_{t_i}} \|G(x_0, x_1, t) - \ell(x_t \mid x_0, x_1, x_{t_i}, t)\|^2, \tag{14}$$

*the same $G(\cdot, \cdot, t)$ minimises $L_{\text{reg}}$ for $U$-a.e. $t$.*

*Proof of Theorem 2.2.* The proof of the first part of the theorem follows similar arguments as in Theorem 2.1. Let $G(\cdot, \cdot, t)$ be any function satisfying the constraint. Consider the following joint distribution on $\mathcal{X} \times \mathcal{X}$:

$$\Pi(dx_t, dx_t') = \left(\ell(\cdot \mid \cdot, \cdot, \cdot, t), G(\cdot, \cdot, t)\right)_{\#}\pi_{t_i}. \tag{18}$$

The marginals of $\Pi$ over the first and second components are the a.c. distributions $\ell(\cdot \mid \cdot, \cdot, \cdot, t)_{\#}\pi_{t_i}$ and $q_t$, respectively. As in Theorem 2.1, the distribution $\Pi$ is a Kantorovich plan between these two marginals whose cost is

$$\int \|x_t' - x_t\|^2 \, d\Pi(x_t, x_t') = \mathbb{E}_{(x_0, x_{t_i}, x_1) \sim \pi_{t_i}} \|G(x_0, x_1, t) - \ell(x_t \mid x_0, x_1, x_{t_i}, t)\|^2, \tag{19}$$

i.e., the regulariser in (11). This cost has a unique minimiser over all transport maps $\Pi$ (up to a.e. equality on the supports) because the first marginal is a.c., and the minimiser has $\Pi$ deterministic over the second component given the first, i.e., $\Pi = (\text{Id}, T)_{\#}\mu_t$ for some function $T : \mathcal{X} \to \mathcal{X}$, where $\mu_t = \ell(\cdot \mid \cdot, \cdot, \cdot, t)_{\#}\pi_{t_i}$. This minimum is indeed uniquely achieved by

$$G(x_0, x_1, t) = T \circ \ell(x_t \mid x_0, x_1, x_{t_i}, t), \quad (x_0, x_{t_i}, x_1) \sim \pi_{t_i}, \tag{20}$$

which proves existence and uniqueness on the support of $\pi$. Since the argument is pointwise in $t$, integrating with respect to any $U$ on $[0, 1]$ (as in (11)) preserves optimality for $U$-a.e. $t$. $\square$

## B  KNOT DISTRIBUTION

We let $t \in [0, t_{\max}]$, ensure that the number of marginals, $K$, is a multiple of three, and set

$$\tilde{t} = 3\frac{t}{t_{\max}} - 1.5. \tag{21}$$

We then partition $\tilde{t}$ into three equally sized intervals,

$$\mathcal{I}_1 = \tilde{t}_{1:K/3}, \quad \mathcal{I}_2 = \tilde{t}_{K/3+1:2K/3} \quad \mathcal{I}_3 = \tilde{t}_{2K/3+1:K},$$

and sample $X$ and $Y$ coordinates as follows:

$$\begin{bmatrix} X(t) \\ Y(t) \end{bmatrix} \sim \mathcal{N}\left(\begin{bmatrix} \mu_X(t) \\ \mu_Y(t) \end{bmatrix}, \sigma^2 I_2\right) \tag{22}$$

with $\sigma = 0.1$ and

$$\mu_X(t) = \begin{cases} 3(t + 0.5) & \text{if } t \in \mathcal{I}_1 \\ \cos(2\pi(t - 0.75)) & \text{if } t \in \mathcal{I}_2 \\ 3(t - 0.5) & \text{if } t \in \mathcal{I}_3 \end{cases} \tag{23a}$$

$$\mu_Y(t) = \begin{cases} -0.5\tanh(5(t + 1)) + 0.5 & \text{if } t \in \mathcal{I}_1 \\ \sin(2\pi(t - 0.75)) & \text{if } t \in \mathcal{I}_2 \\ 0.5\tanh(5(t - 1)) + 0.5 & \text{if } t \in \mathcal{I}_3 \end{cases} \tag{23b}$$

Finally, we divide $t$ by three and collect ten samples from the bivariate distribution of $(X(t), Y(t))$ at each of the $K$ time stamps.

## C  PREPROCESSING THE SPATIAL TRANSCRIPTOMICS DATA

A sequence of spatial transcriptomics tissue sections (e.g., in Fig. 7) is not naturally aligned since each slice is cut separately, for instance, at slightly different orientations. Small physical distortions occur during sectioning, staining, storing (the tissues are typically frozen post-sectioning), and imaging, causing stretching, folding, or rotation relative to neighbouring sections. In fact, practitioners manually place the tissues on a slide. As a result, the spatial coordinates reported for each section exist in their own local coordinate system and cannot be directly mapped to the spots of other sections without an alignment preprocessing step. We are particularly interested in aligning the spatial coordinates of the spots (the spots, or features, contain the RNA expression).

We aligned the spots by first visually aligning the raw H&E-stained images shown in Fig. 7 using the BigWarp plugin (Bogovic et al., 2016) in Fiji (Schindelin et al., 2012), an open-source biological-image analysis software. After downscaling the H&E-stained images by a factor of 1/10, we pairwise aligned all images to the fourth section (Fig. 7d). To align a pair of images in BigWarp one manually chooses landmarks which the software then utilises to nonlinearly warp the source image to the target image (again, Fig. 7d is the target image in all image pairs). The above accurately aligned the pixel coordinates of the source image to the target, which was clear from visual inspection.

Finally, we transformed the spot coordinates in the source section to the coordinate system of the target section. In particular, the nonlinear transformation used above (thin plate spline transformation) provided a mapping from the source pixel coordinates to the target pixel coordinate system. We found that feeding these maps to a `scipy.interpolate.RegularGridInterpolator` object to transform the source spot coordinates accurately aligns the spot coordinates with the warped source images. The result of our alignment preprocessing is shown in Fig. 5.

## D  ADDITIONAL EXPERIMENTAL DETAILS AND RESULTS

### D.1  KNOT EXPERIMENT

Our proposed method is trained using two-layer MLPs with 128 hidden units and ELU activation function for both $f_\phi$ and $D_\gamma$. We use $\lambda = 1$ in (8), the linear reference regulariser from (9), and we

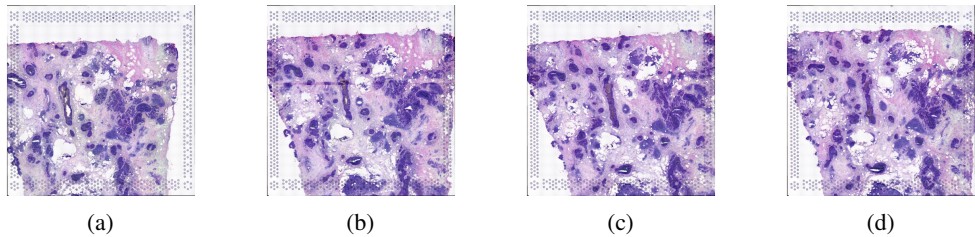

|  (a)  |  (b)  |  (c)  |  (d)  |

Figure 7: The raw H&E-stained images of the four tissue sections provided by Mo et al. (2024).

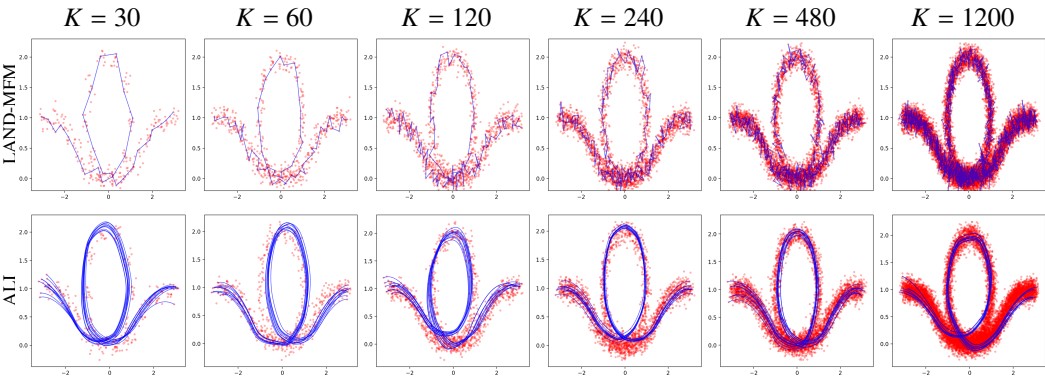

Figure 8: Comparison between learnt time-dependent LAND-based interpolants used in OT-MFM, and the ALI interpolants for different number $K$ of marginal distributions on the knot dataset (§4.1).

train the interpolant for $50,000$ epochs with a batch size of 128. The nets $f_\phi$ and $D_\gamma$ are optimised using separate Adam optimisers with a learning rate of $10^{-3}$. The flow matching networks are parameterised as two-layer MLPs with 32 hidden units and SELU activations in every experiment, and the Adam optimiser (Kingma and Ba, 2015) is used for training. OT-ALI-CFM, OT-CFM and OT-MMFM were trained for 30000 epochs with a learning rate of $10^{-4}$. See below for OT-MFM details.

**OT-MFM hyperparameters with time-independent LAND metric**  We train OT-MFM for 10, $1,000$, and $40,000$ epochs with Adam optimiser and learning rate $10^{-4}$ for both the interpolant and flow matching networks. We experimented with other learning rates; however, we observed no difference or a degraded performance. We kept the default hyperparameters from the MFM codebase (Kapuśniak et al., 2024) with the LAND metric. Following the net parameter choices for ALI on this data, the interpolant network has two hidden layers with 128 units, and the vector field net has two layers and 64 hidden units. All networks have ELU activations. The batch size is set to 128.

**OT-MFM with time-dependent LAND metric**  We additionally ablate the use of LAND metirc in MFM (Kapuśniak et al., 2024) algorithm. Fig. 8 shows how the learnt interpolants for MFM and ALI change as the number of marginal distributions increases. When OT-MFM is equipped with a time-dependent LAND metric, the interpolants are constrained to pass through the marginal samples pointwise. As such, it is unsurprising that the resulting geopath interpolants resemble the piecewise linear and cubic spline interpolants in Fig. 2. As we see, the interpolants, trained using LAND MFM become highly non-smooth, whereas ALI interpolants stay smooth regardless the number of marginals.

As for the other methods in Fig. 2, we trained the vector field for 30,000 iterations with a learning rate of $10^{-4}$, and a two-layered MLP with 32 hidden units per layer and SELU activations.

**OT-MFM with a time-dependent RBF metric**  Beyond the setting above, we also attempted to run MFM on the knot data with the time-dependent RBF metric, as it is implemented in the publications' associated code (see the single-cell trajectory inference section in Kapuśniak et al. (2024) when the dimensions are 50 or 100D).

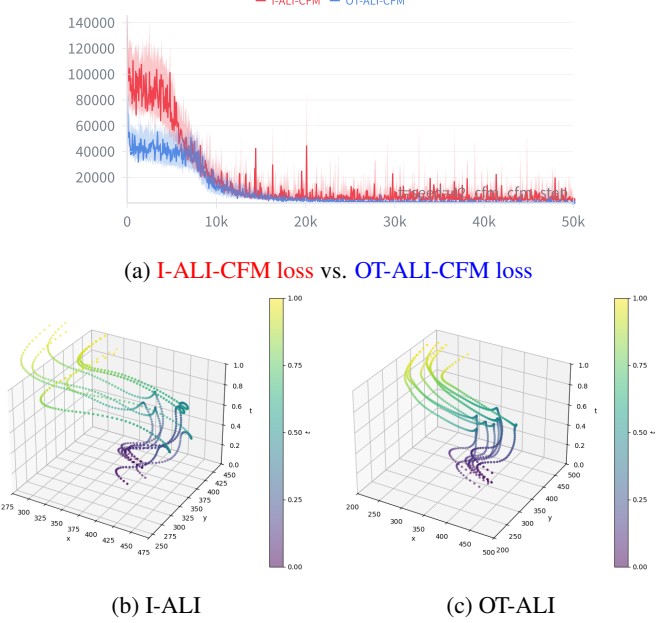

(a) I-ALI-CFM loss vs. OT-ALI-CFM loss

(b) I-ALI                                    (c) OT-ALI

Figure 9: Comparison of I-ALI-CFM and OT-ALI-CFM on the cell tracking data. As expected, the CFM losses in (a) show the increased variance in the CFM objective when using the independent coupling (red) compared to the OT coupling (blue). The effect of the increased variance is shown in (b) and (c), where the I-ALI trajectories overlap to a greater extent than the OT-ALI trajectories.

As outlined in §4.1, there are $K = 1200$ marginal distributions with ten samples from each distribution. Before we can train the MFM interpolant (geopath) network, we therefore need to train 1199 RBF networks via gradient descent.

We optimised the hyperparameters in the RBF training in order to jointly optimise training time and metric loss, without getting NaN losses when training the 1199 networks. Each network is, by the construction of the experiment and the time-dependent RBF implementation, fitted to 20 samples.

With five clusters, hundreds of RBF nets had NaN losses independent of learning rates. This happened due to numerical issues when computing the empirical variance of the cluster-assigned points (line 76 in `https://github.com/kkapusniak/metric-flow-matching/blob/main/mfm/geo_metrics/rbf.py`). We found that two clusters did not result in NaN losses, and then used 50 training epochs per RBF net with the default learning rate, ensuring converging loss curves. The training time—to fit the metrics, no training of the interpolant or flow nets—was around 400 minutes, i.e., 6.5 hours, on an NVIDIA GeForce RTX 3080 GPU. Recall that this is 2D data.

### D.2  CELL TRACKING EXPERIMENT

The data can be found on this website, or explicitly via this link: `https://data.celltrackingchallenge.net/training-datasets/PhC-C2DH-U373.zip`. We use the frames available in folder `01` and the corresponding segmentation masks from folder `01_ST/SEG`. We used the segmentation masks for the cell with label four.

**ALI hyperparamters.**   Both the interpolant and the discriminator are parameterised as 2-layered MLPs with ELU activations and 256 hidden units. We train ALI for $70,000$ epochs. During training, we smooth the time-input to the generator by adding Gaussian noise with standard deviation $10^{-3}$. Both networks are trained with learning rates of $10^{-4}$ and separate Adam optimisers.

**OT-MFM with a time-independent LAND metric.**   For the cell tracking experiments OT-MFM is trained for $40,000$ epochs with learning rate $10^{-4}$ for both interpolant and flow matching networks. The weights of both networks are optimised with Adam. We use $\gamma = 0.4$ and $0.9$ performing similarly, while leaving other hyperparameters of the LAND metric default. Training for longer also does not

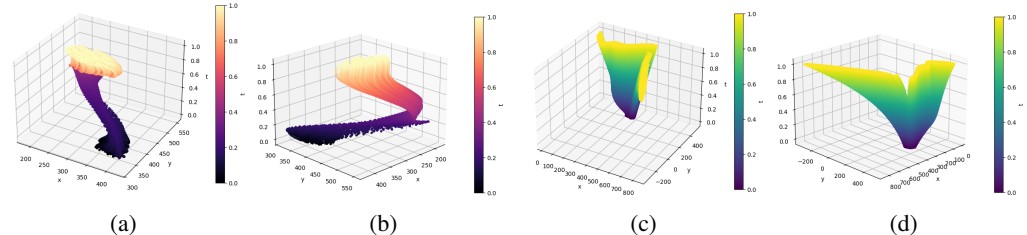

Figure 10: Different 3D views of all time-independent OT-MFM interpolants are shown in **(a)** and **(b)**, while in **(c)** and **(d)** we show the 3D visualisations of the marginalised OT-MFM trajectories. In **(b)** we observe that the interpolants start to bifurcate around $t \approx 0.5$ and then abruptly change direction in order to satisfy the end-marginal constraint. We believe the bifurcation happens due to the guidance of the metric that was fitted on all training samples. Crucially, the bifurcation in the interpolation paths causes part of the vector field to diverge, which is seen in **(c)** and **(d)**. Note the extreme range of $x$-values in the 3D plots in **(c)** and **(d)** when mapping them to the scatter plots in Fig. 4.

yield better results with these settings. Batch size is set to 128. The neural networks have 128 hidden units and 3 hidden layers.

**OT-MFM with a time-dependent LAND metric.** For these experiments we trained both networks (same architecture as for other OT-MFM experiments) for 75000 epochs and learning rate $10^{-4}$ (Adam); best performing hyperparameters were $\gamma = 0.2$ and other LAND hyperparameters set default. While it would be possible to run OT every possible batch of $(x_{t_i}, x_{t_{i+1}})$ during training, we instead precompute OT maps between the full successive pairs of marginals before training once and reuse them, as the former option was too computationally expensive.

**I-ALI-CFM vs. OT-ALI-CFM** In Fig. 9 we compare I-ALI-CFM and OT-ALI-CFM. As expected, the CFM loss curves exhibit higher variance for I-ALI-CFM, while the resulting I-ALI trajectories overlap to a greater extent than the OT-ALI trajectories.

### D.3 SCRNA-SEQ EXPERIMENTS

For single-cell experiments, we use the dataset available on `https://data.mendeley.com/datasets/hhny5ff7yj/1`, the EB data used in Tong et al. (2020) can be found on `https://github.com/KrishnaswamyLab/TrajectoryNet/blob/master/data/eb_velocity_v5.npz`. Following Tong et al. (2024), we additionally whiten the data in the 5D experiments.

**Hyperparameters for ALI-CFM.** We parameterise learnable correction $f_\phi$ and $D_\gamma$ as two-layered MLPs with 64 (for the 5D experiments) or 1024 (for the 50 or 100D experiments) hidden units in each layer with ELU activations. In accordance with Kapuśniak et al. (2024), we use a three-layered MLP with SELU activations to model $u_t^\theta$. The number of hidden units in the layers of $u_t^\theta$ is chosen to be either 64 or 1024, depending on the data dimensions, as for $f_\phi$ and $D_\gamma$ above. We find that normalising the data and adding small noise ($0.01 \cdot \epsilon$, $\epsilon \sim \mathcal{N}(0, 1)$) to the time input of $G_\phi$ helps when training the GAN, as it makes the interpolant smoother with respect to time input. At inference, we push all the samples from marginal $i - 1$ to the time associated with the held-out marginal $t_i$, as in Tong et al. (2024).

For all of the single-cell datasets we pretrain the interpolant for $2,000$ steps and then train it using adversarial losses for $70,000$ steps. We empirically found that the best results on single-cell data are obtained by using R3GAN Huang et al. (2024) with the second-derivative regulariser. We adjust the scale of the regulariser term in such a way that it has the same value range as the generator loss. We use learning rate $10^{-3}$ for pretraining, $5 \cdot 10^{-5}$ for both discriminator and generator for adversarial training and $10^{-3}$ for the OT-CFM training. We find that OT coupling is important to achieve better results. We use multi-marginal OT coupling in all our experiments. We also investigate the alternatives for the $L_2$-norm. Following Kapuśniak et al. (2024), we investigate substituting the $L_2$-norm with the "LAND" metric in all of the previously discussed regularisers. Given the dataset

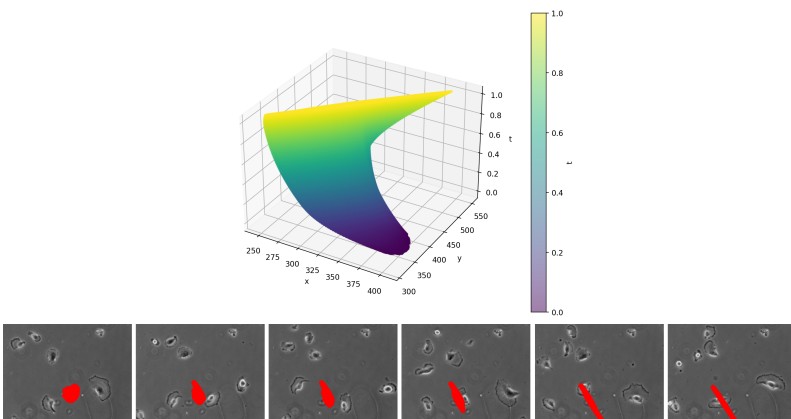

Figure 11: **Top:** The time-dependent OT-MFM marginalised vector field in 3D on the cell tracking data. **Bottom:** the trajectories overlayed on the a subset of the microscopy image frames (see Fig. 4).

$\mathcal{D} = \{(x_{t_i}, t_i)\}_{i=1}^N$ and $\epsilon > 0$ we let $(x_t, t) \mapsto g_{\text{LAND}}(x_t, t) \equiv G_\epsilon(x_t, t) = (\text{diag}(h(x_t, t)) + \epsilon I)^{-1}$, where

$$h_\alpha(x_t, t) = \sum_{(x_s, s) \in \mathcal{D}} (x_t - x_s)^2 \exp\left(-\frac{\|x_t - x_s\|^2}{\gamma_1}\right) \exp\left(-\frac{\|t - s\|^2}{\gamma_2}\right), \quad 1 \leq \alpha \leq d, \qquad (24)$$

with $d$ being the dimension of $x$ and $\gamma_1$ and $\gamma_2$ being the kernel sizes. Using "LAND" improved the results. We conduct our final experiments on single-cell data using the regulariser detailed in Equation (15) with the "LAND" metric.

**Reproduced results.** In order to fairly compare our method to the existing ones, we rerun all the baselines on our hardware. The reported scores I/OT-MFM and I/OT-CFM are obtained by running the code provided by Kapuśniak et al. (2024) without any changes made to the code. For I-MMFM and OT-MMFM (Rohbeck et al., 2025), we implemented the cubic splines interpolants without the Gaussian probability path, to align with the other considered methods. We only ran I/OT-MMFM on the 5D experiments, because this method poorly scales to the high dimensions.

### D.3.1 ABLATION STUDY FOR THE DIFFERENT REGULARISATION TERMS

We conduct an ablation study on different dimensions of the CITE-seq data, ablating regularisers (§2.3), regulariser coefficients, and the effect of these on EMD, computed on the left out time steps, and path energy, which quantifies the straightness of the trajectories (higher numbers, less straight paths) and is computed in the following way:

$$\text{path energy} = \mathbb{E}\left[\int_0^1 \|u_t^\theta(x_t)\|^2 dt \,\bigg|\, x_0 \sim q_0\right]. \qquad (25)$$

The results are shared in Tables 4, 5, 6.

Interestingly, we see that increasing the regularising coefficient, combined with the piecewise linear regulariser, makes the trajectories less smooth. We argue that this is because the piecewise linearity forces interpolants to pass through the marginal samples pointwise, akin to OT-CFM, MFMM and MFM with a time-dependent metric, and so the trajectories become less straight.

Meanwhile, a stronger regularising weight combined with the other regularisers in general straightens the interpolants (decreases the path energy). This is expected as they directly penalise curvature.

### D.4 SPATIAL TRANSCRIPTOMICS EXPERIMENT

In Fig. 6, we visualise some of the resulting tumour coordinate inference results.

When marginalising the vector fields during inference of tumour coordinates at the held-out time $t_i$, we push the samples from the observed $q_{t_{i-1}}$ to $t = t_i$ using 101 integration steps and the `dopri5`

Table 4: EDM ($\downarrow$) for the learnt interpolant (Interpolant) and the marginalised vector field (CFM), as well as path energy ($\downarrow$) for the marginalised vector field (CFM path energy), computed on the left out time steps for Cite 50D and piecewise linear regulariser

| | Left out time step $t = 1$ | | | Left out time step $t = 2$ | | |
|---|---|---|---|---|---|---|
| $\lambda$ | Interpolant | CFM | CFM path energy | Interpolant | CFM | CFM path energy |
| 0.01 | 46.96 | 47.29 | 300.01 | 38.12 | 42.17 | 358.98 |
| 0.05 | 41.11 | 41.07 | 697.19 | 37.70 | 41.17 | 385.26 |
| 0.10 | 40.76 | 41.19 | 1243.33 | 37.28 | 40.51 | 551.10 |
| 0.20 | 41.21 | 42.20 | 2979.58 | 38.28 | 41.75 | 238.59 |
| 0.50 | 41.82 | 40.93 | 5666.17 | 37.00 | 39.48 | 268.40 |
| 5 | 42.23 | 42.59 | 36264.89 | 35.93 | 41.70 | 6297.56 |
| 10 | 41.38 | 43.81 | 87052.14 | 35.57 | 43.06 | 8484.26 |

Table 5: EDM ($\downarrow$) for the learnt interpolant (Interpolant) and the marginalised vector field (CFM), as well as path energy ($\downarrow$) for the marginalised vector field (CFM path energy), computed on the left out time steps for Cite 50D and 'the norm of the second derivative' regulariser

| | Left out time step $t = 1$ | | | Left out time step $t = 2$ | | |
|---|---|---|---|---|---|---|
| $\lambda$ | Interpolant | CFM | CFM path energy | Interpolant | CFM | CFM path energy |
| 1.00E-06 | 41.50 | 42.17 | 12.30 | 39.63 | 45.66 | 7.58 |
| 5.00E-07 | 41.48 | 42.14 | 12.00 | 39.55 | 45.51 | 7.55 |
| 1.00E-07 | 41.44 | 42.06 | 12.56 | 39.35 | 45.23 | 7.68 |
| 5.00E-08 | 41.38 | 42.23 | 13.91 | 39.31 | 44.99 | 8.10 |
| 1.00E-08 | 40.75 | 41.59 | 26.00 | 39.06 | 41.61 | 13.21 |
| 5.00E-09 | 41.39 | 42.24 | 41.98 | 39.04 | 40.14 | 18.51 |
| 1.00E-09 | 42.89 | 43.81 | 56.39 | 41.24 | 40.04 | 39.65 |

ODE solver. For all methods, the dataset is normalised to make the coordinates be in $[0, 1]$. After training the CFMs, the marginalised CFM trajectories are denormalised and compared with the unnormalised true data.

**ALI hyperparameters.** We train $G_\phi$ and $D_\gamma$ for $70,000$ epochs with the piecewise linear reference in (11), and $\lambda = 10$. We use a batch size of 256. Additionally, we pretrain $G_\phi$ for 1000 epochs using a simple $L_2$ regression objective between $G_\phi$ and samples from the observed $q_{t_i}$. During training, we add a small Gaussian noise variable with standard deviation parameter 0.001 to the time input of the generator. The learning rates in the adversarial training were $10^{-5}$ for both nets, and we used the Adam optimiser with default hyperparameters.

**CFM hyperparameters.** ALI-CFM is trained using a learning rate of $10^{-3}$. Although we have tested multiple learning rates for OT-CFM and OT-MMFM, the best performance is attained using $10^{-3}$ for OT-CFM and $10^{-4}$. All nets are optimised using the Adam optimiser.

**OT-MFM hyperparameters.** We train OT-MFM for $40,000$ epochs with learning rate $10^{-4}, 2 \cdot 10^{-4}$ and Adam for the interpolant and flow matching network respectively. We note that we do not report results with the multi-marginal version of OT-MFM using Eq. (20) of Kapuśniak et al. (2024); in our preliminary tests, this version exhibited worse results than the standard OT-MFM.

Batch size is set to 128. We use the LAND metric as this is the recommended one for low-dimensional data in Kapuśniak et al. (2024). We set the hyperparameters of the metric to $\gamma = 0.2$ for ST data and 0.4 for cell tracking data (other values perform worse, respectively); $\rho = 5 \cdot 10^{-4}$ for ST data seems to perform better than the default; the remaining hyperparameters are set to MFM defaults. The neural networks have 128 hidden units and 3 hidden layers.

Table 6: EDM (↓) for the learnt interpolant (Interpolant) and the marginalised vector field (CFM), as well as path energy (↓) for the marginalised vector field (CFM path energy), computed on the left out time steps for Cite 100D and 'the norm of the second derivative' regulariser

| | Left out time step $t = 1$ | | | Left out time step $t = 2$ | | |
|---|---|---|---|---|---|---|
| $\lambda$ | Interpolant | CFM | CFM path energy | Interpolant | CFM | CFM path energy |
| 1.00E-06 | 47.00 | 49.89 | 6.90 | 44.78 | 51.12 | 6.90 |
| 5.00E-07 | 47.03 | 49.37 | 6.66 | 44.88 | 50.92 | 6.60 |
| 1.00E-07 | 47.03 | 49.65 | 10.02 | 44.80 | 50.99 | 7.09 |
| 5.00E-08 | 46.87 | 49.04 | 11.25 | 44.77 | 51.19 | 11.86 |
| 1.00E-08 | 46.66 | 48.36 | 25.43 | 44.65 | 49.98 | 31.18 |
| 5.00E-09 | 46.52 | 47.84 | 30.93 | 44.74 | 50.00 | 37.22 |
| 1.00E-09 | 46.97 | 47.92 | 55.48 | 45.97 | 49.60 | 28.22 |

