# OpenReview forum: "Multi-Marginal Flow Matching with Adversarially Learnt Interpolants"
_ICLR.cc/2026/Conference — ICLR 2026 Poster_

### Official Review · Reviewer_YnZ4 · 2025-10-25

**Soundness:** 1
**Presentation:** 4
**Contribution:** 2
**Rating:** 2
**Confidence:** 4

**Summary:**

The paper proposes Adversarially Learned Interpolants: a GAN-style objective that directly matches intermediate-time marginals to a learnable interpolant. The full ALI-CFM pipeline first learns adversarially (with novel regularisers for uniqueness/smoothness), then regresses a time-dependent vector field via a standard CFM loss. The generator and discriminator are time-dependent, and no explicit metric is specified.

The method is evaluated against existing flow matching baselines on several tasks: a synthetic 2D knot dataset , a real-world cell tracking dataset, a standard single-cell trajectory inference benchmark , and a novel task of inferring tumour coordinates from spatial transcriptomics data.

**Strengths:**

1.  **Novel Regularisers** - the linear-reference and curvature penalties are well-motivated for stabilising adversarial training and encouraging uniqueness of interpolants
2.  **New Benchmark Task** - Tumour coordinate inference with spatial transcriptomics is a relevant multi-marginal benchmark; the paper positions ALI-CFM credibly there and reports reproducible settings
3.  **Problem Importance**  - The paper proposes an elegant approach to well motivated problem

**Weaknesses:**

1. **MFM Baseline Comparison**
   The paper’s central argument is that MFM is *“not suitable when the data geometry varies with time”* because their metric is *“time-independent.”* However, the statement *“their metric is time-independent, which contrasts with our generator and discriminator networks which are both time-dependent”* is somewhat misleading.  The authors explicitly state that their MFM implementation *“follows the setup … where the metric is inferred from all the data.”* This is a **specific, time-independent implementation choice**, not an inherent limitation of the MFM framework.  In fact, the LAND metric can be defined **locally in time** (e.g., using only data from the nearest time points, as done in MFM’s single-cell setup). Such a *time-localized* definition would make the metric effectively time-dependent and would likely yield more accurate MFM interpolants, especially in Figure 2.  This would basically be equivalent to time dependance learnt through Adversarial training as its the only granularity of information that we posses.

2. **Novelty Claim**
   The paper frames its novelty as introducing a time-dependent adversarial approach that overcomes previous work time-independence. However, a time-conditioned discriminator (as used in ALI) and a time-localized LAND metric (as in a correctly implemented MFM) are conceptually similar—both use data from neighboring time points to define geometry along the interpolant.   If a fair, time-localized MFM implementation performs competitively, the core premise of the paper weakens: the proposed method may represent an alternative, but not fundamentally new, way of learning time-dependent interpolants—albeit through a more complex, GAN-based training procedure.

3. **Minor: Attribution of the Interpolant Form**
   The interpolant parameterization in Equation (5), $G_{\phi}(x_{0},x_{1},t)=(1-t)x_{0}+tx_{1}+t(1-t)f_{\phi}(\cdot)$,  is a standard form introduced in prior works (e.g., Neklyudov et al., 2024; Kapuśniak et al., 2024).  While these papers are cited in the related work section, the equation itself appears without a direct attribution, which could unintentionally suggest it is novel.

**Questions:**

1. **Baseline fairness and time-localization of LAND.**
   Did your baselines using the LAND metric employ the *adjacent-time* constraint as in the original MFM implementation?
   If not, could you please report results using the time-step–localized (i.e., nearest-time) implementation of LAND as discussed in the Weaknesses section, and reflect these results in Figures 2, 4, 7, and Table 3?

2. **Validity of the “geometry varies with time” claim.**
   If the above concern holds, the statement that LAND/RBF-based and prior methods are *“not suitable when the data geometry varies with time”* may not be accurate. Is there any additional theoretical or empirical reason that supports this claim?

3. **Novelty.**
   If the above concern is valid and a fair, time-localized MFM performs well, what do the authors consider to be the primary novelty of ALI-CFM beyond offering an alternative formulation of a time-conditioned interpolant?

---

> ### Author Response · Authors · 2025-11-21
>
> We thank the reviewer for the insightful comments regarding the comparison with MFM. We would like to address the claims about comparison between our method and the baselines, as well as other comments regarding the complexity of our methods.
>
> ### **Equivalence between time dependence as modeled by our method and MFM**
>
> Although it is true that MFM employs a time-dependent *RBF metric* in the single-cell trajectory inference experiment, **the LAND metric used is not time-dependent**, as is clear both from the paper and from the released code. That is, a time-dependent LAND metric is not available, and there is no discussion of such a metric in the publication. We see multiple possibilities for defining such a metric, including fitting it to two successive time points (as is done in the RBF metric) or a ‘soft’ variant of this that fits the metric in a joint position-time space — but **studying such generalisations of MFM is not the purpose of our work**. In contrast, ALI-CFM naturally handles time dependence by making time a continuous-valued input to both the generator (interpolant) and discriminator models, which allows generalisation between time steps.
>
> Nonetheless, we agree with the reviewer that a comparison with MFM using a time-dependent metric is in order.
>
> ### **MFM baseline comparison**
>
> We show empirically that MFM with a time-dependent metric, as it is implemented in the single-cell experiment, **scales poorly with the number of marginal distributions, $K$,** especially if the number samples per marginal is small. Notably, this is precisely the data setting in the synthetic knot distribution and in the cell tracking data. We have trained an MFM model with a time-dependent RBF on the knot distribution, which has over 1000 discrete time points with 10 data samples per marginal. While faithfully tuning the hyperparameters in the RBF training in order to jointly optimise training time and metric loss, we could not fit a time-dependent RBF metric in under **6.5 hours** (a more detailed experimental protocol is provided below). Note that this is without factoring in any training of $\varphi$ (the correction/geopath network), or the vector field network. In contrast, training ALI-CFM on this data took ~40 minutes in total, without early stopping.
>
> **In summary**, though we agree that it is **conceptually** possible to generalise MFM to this ‘continuous-time’ setting, MFM as it is currently presented does not allow for it, nor is it explained or explored in the MFM paper how to achieve it.
>
> Furthermore, after extensive testing of different hyperparameter choices, no **MFM with a time-dependent RBF metric on the ST data** could outperform its time-independent counterpart (the scores from the time-independent version are the ones in our submission). We believe that this is due to MFM being sensitive to multimodality: If we use too many RBF centres, some of them end up without assigned samples, resulting in NaNs when computing the empirical variance. If we use too few centers, on the other hand, the manifold does not include all the modes, resulting in poor EMD scores. This suggests that an RBF metric fit to adjacent time points is unsuitable for sparse data.
>
> **In-depth training protocol (knot distribution):** As outlined in our submission, there are $K=1200$ marginal distributions with 10 samples from each distribution. Before we can train the geopath network, we therefore need to train 1199 RBF networks via gradient descent. Already here it becomes quite apparent that MFM is not well-suited for handling this data scenario.
>
> We optimised the hyperparameters in the RBF training in order to jointly optimize training time and metric loss, without getting NaN losses when training the 1199 networks. Each network is, by the construction of the experiment and the time-dependent RBF implementation, fitted to 20 samples.
>
> With five clusters, hundreds of RBF nets had NaN losses independent of learning rates. This happened due to numerical issues when computing the empirical variance of the cluster-assigned points (line 76 in rbf.py). We found that two clusters did not result in NaN losses, and then used 50 training epochs per RBF net with the default learning rate. The training time — to fit the metrics, no training of the interpolant or flow nets — was around 400 minutes, i.e., ~6.5 hours, on an NVIDIA GeForce RTX 3080 GPU. Recall that this is 2D data.

---

> > ### Author Response · Authors · 2025-11-21
> >
> > ### **Novelty claim**
> >
> > Given the above, we respectfully disagree that a *time-conditioned discriminator and a time-localised metric are conceptually similar*. Conversely, we use the above examples to highlight the distinction between the two methods, and to determine that *a fair, time-localized MFM implementation* **does not** *perform competitively* with our ALI-CFM, which uses a time-conditioned discriminiator. This also demonstrates that adversarial training does not add complexity, but can instead **highly reduce runtime**. Finally, we stress that our adversarial approach to learning time-dependent dynamics in flow matching is not only *fundamentally new,* but can also be empirically superior in multiple interesting data settings.
> >
> > ### **Minor: Attribution of the interpolant form**
> >
> > You are absolutely correct to point this out, we will add a reference to the two works directly after Equation (5), and not only in the related work. Thank you for bringing this up.
> >
> > ### Questions
> >
> > **Q1 and Q3:** We kindly ask the reviewer to acknowledge Q1 and Q3 as resolved through our responses above.
> >
> > **Q2:** *Is there any additional theoretical or empirical reason that supports* the claim that *LAND/RBF-based and prior methods are “not suitable when the data geometry varies with time”.*
> >
> > Yes, encouraged by your review, we have trained MFM with time-dependent metrics on our datasets and have above provided clear empirical support for this claim.
> >
> > **We are sincerely grateful for the reviewer’s comments. The new baseline experiments more faithfully determine the distinction between our method and MFM with a time-dependent metric, as well as emphasise the relevance of ALI-CFM when $K$ is large.**
> >
> > **We kindly ask the reviewer to revise their assessment in light of the above clarifications of our contribution. We are happy to address any further questions.**

---

> > > ### Comment · Reviewer_YnZ4 · 2025-11-27
> > > **Response to Author's rebuttals**
> > >
> > > I thank the authors for their detailed response, particularly regarding the scalability issues of RBF metrics in high-marginal settings. I acknowledge the empirical evidence provided regarding the computational cost of training ~1200 RBF networks for the knot experiment. This is a valid argument for the efficiency of ALI-CFM in specific high-$K$ regimes.
> > >
> > > However, I must push back strongly on the characterization of the MFM baseline and the resulting narrative regarding "time-dependence."
> > >
> > > **1. Factual Correction on MFM and Time-Dependence**
> > > The authors state in the rebuttal: *"the LAND metric used is not time-dependent, as is clear both from the paper and from the released code... fitting it to two successive time points ... is not the purpose of our work."*
> > >
> > >  This is factually incorrect. The standard implementation of MFM (and optimal transport-based flow matching in general) operates on pairs of marginals.
> > > * **MFM Paper (Section 3.1):** Explicitly states: *"in practice $\mathcal{D}$ is constructed concatenating samples from both the source and target distributions."*
> > > * **MFM Codebase:** In `https://github.com/kkapusniak/metric-flow-matching/blob/main/mfm/flow_matchers/geopath_net_train.py` (lines 74-98), the data $\mathcal{D}$ (variable `samples`) used to define the metric is constructed from the specific source/target pair being matched.
> > >
> > > Therefore, if one applies MFM to a multi-marginal problem by matching sequential pairs $(q_{t_i}, q_{t_{i+1}})$, the metric is inherently time-dependent (piecewise), as it is constructed solely from the data at $t_i$ and $t_{i+1}$. To define a "global," time-independent metric across all $K$ marginals (as done in Figure 2 of your submission) is actually a deviation from the standard pairwise training setup of flow matching.
> > >
> > > **2. Misleading Comparisons and Narrative**
> > > Because of the above, Figure 2 is misleading. It compares your method (which sees local geometry) against an MFM implementation forced to use a global metric derived from *all* data, rather than the standard pairwise implementation.
> > >
> > > Furthermore, the manuscript currently claims: *"However, their metric is time-independent, which contrasts with our generator and discriminator networks, both of which are time-dependent."*
> > > Based on the evidence above—and your own admission in the rebuttal that RBF can be time-dependent—this statement is false. The novelty of ALI-CFM is not *that* it is time-dependent, but perhaps *how* it handles time-dependence (via adversarial learning vs. metric construction).
> > >
> > > **3. The Knot Experiment and LAND**
> > > While I accept that training 1200 RBF networks is computationally expensive, the LAND metric does not require training a network (it uses the raw data points). Thus, the scalability argument used to dismiss the RBF baseline does not apply to LAND. There is no computational barrier to running the Knot experiment with a standard, pairwise (time-dependent) LAND metric to provide a fair comparison.
> > >
> > > **Conclusion**
> > > I see merit in the proposed method, particularly regarding scalability in high-$K$ regimes where constructing metrics becomes tedious. However, I cannot recommend acceptance while the paper relies on a false dichotomy regarding time-dependence.
> > >
> > > I am willing to reconsider my score if the following two actions are taken:
> > >
> > > 1.  **Fair Baselines:** Re-run all examples where the LAND metric is used (specifically the Knot experiment and relevant figures/tables) using the standard, pairwise (time-dependent) formulation of MFM. Since LAND requires no training, the computational overhead is minimal compared to the RBF scalability issues you noted.
> > > 2.  **Corrected Narrative:** Update the manuscript to remove the claim that previous methods are inherently "time-independent" or unsuitable for time-varying geometry. Explicitly acknowledge that MFM is locally time-dependent when trained pairwise, and reframe the novelty around the specific advantages of the adversarial formulation (e.g., efficiency) rather than the existence of time-dependence itself.

---

> > > > ### Author Response · Authors · 2025-12-03
> > > >
> > > > We would like to thank the reviewer for their thorough reading of our paper and for the scrutiny to our work. The comments helped us to improve our method and get a better understanding of the baseline. We strongly believe that we have now addressed all of the reviewer’s concerns.
> > > >
> > > > In the rebuttal below we provide a detailed response showing that our claims about the advantages of ALI-CFM hold even with the corrections made.
> > > >
> > > > ### **On the discrepancy with [Kapuśniak et al.] and our changes**
> > > >
> > > > The main point of discrepancy, as we see it, occurred with us not using the time-dependent version of the LAND metric in MFM [Kapuśniak et al.]. After the reviewer pointed us to the original code and showing how the time-dependent LAND metric should be used, we updated the results of the experiments incorporating the runs with the time-dependent LAND metric.
> > > >
> > > > To facilitate fair comparison with the baselines we have run the following experiments:
> > > >
> > > > 1. We reran the knot experiment with MFM, with the LAND metric computed between every two consecutive marginals. We include the pseudocode snippets for comparison between the original use of LAND metric and the corrected one.
> > > >
> > > >     ```python
> > > >     # Original way of computing LAND metric
> > > >     ...
> > > >     t = sample_timestep()
> > > >     x_t, u_t = compute_position_and_velocity(..., OT=True)
> > > >
> > > >     # we concatenate all the samples into one tensor
> > > >     metric_samples = concatenate(samples)
> > > >     # and use it to compute the LAND metric
> > > >     M_dd = land_metric_tensor(x_t, metric_samples)
> > > >
> > > >     velocity = sum(sqrt((u_t**2 * M_dd), dim=-1))
> > > >     loss = mean(velocity**2)
> > > >     ```
> > > >
> > > >     ```python
> > > >     # Corrected way of computing LAND metric
> > > >     ...
> > > >     loss = 0
> > > >
> > > >     for i in [1, ..., K - 1]:
> > > >     		# sample t \in [t_i, t_{i+1}]
> > > >     		t = sample_timestep(t_i, t_{i+1})
> > > >     		# get x_t and u_t using t, t_i, t_{i+1} and OT coupled x_{t_i} and x_{t_{i+1}}
> > > >     		x_t, u_t = compute_position_and_velocity()
> > > >
> > > >     		# we use only the samples from the neighbouring marginals
> > > >     		metric_samples = concatenate([samples[t_i], samples[t_{i+1}]])
> > > >     		M_dd = compute_land_metric(x_t, metric_samples)
> > > >
> > > >     		velocity = sum(sqrt((u_t**2 * M_dd), dim=-1))
> > > >     		loss += mean(velocity**2) / (K - 1)
> > > >     ```
> > > >
> > > >     The updated results are now included in Figure 2 of the paper.
> > > >
> > > > 2. We reran the cell tracking experiment with the time-dependent LAND metric and have added the results in Fig. 10, together with implementation details in §4.1 and §4.2. Aligned with the results for the knot data, the resulting interpolants unsurprisingly exhibit many kinks that cause the learnt vector field to produce diverging trajectories. This is unsurprising since the time-dependent metric forces the OT-MFM interpolants to pass through the marginal samples pointwise, and so their behaviour resembles that of the other piecewise interpolation methods.
> > > >
> > > > Note that **the results for all other experiments (spatial transcriptomics and single-cell) are unaffected**: Single cell experiments were run directly using [Kapuśniak et al.]’s published code using the RBF metric, while spatial transcriptomics data has only 3 time steps and the middle one is left out during training, meaning that the LAND metric could be computed only using the time steps 1 and 3, which is already done in our original experiments.

---

> > > > > ### Author Response · Authors · 2025-12-03
> > > > >
> > > > > ### **This does not change our main conclusions (+ new experiment in support)**
> > > > >
> > > > > Even with the changed computation of the LAND metric, our main conclusions regarding the time dependence hold: **Piecewise methods, including the time-dependent LAND metric, fail to properly capture the global geometry of the data**.
> > > > >
> > > > > To further support this claim, **we conduct an additional experiment showing how the learnt interpolant changes with the increase in the number of marginal distributions**. From this experiment it is quite evident that methods relying on piecewise interpolation, with or without a metric capturing time-local geometry, scale poorly with the number of intermediate marginals; in contrast, our proposed method is capable of producing smooth interpolants even as the number of marginals grows. The results are presented in Figure 7.
> > > > >
> > > > > We provide the following explanation for why adversarial training is capable of better capturing the underlining global geometry, while MFM is not.
> > > > >
> > > > > - First, piecewise methods extensively rely on the points from consecutive marginal distributions coming in the proper order. Marginals in the knot distribution and the cell tracking data are noisy, which results in the pointwise interpolations of marginals being highly inaccurate and non-smooth. In contrast, adversarially trained interpolants are trained as whole trajectories. Therefore, they are capable of learning smooth approximations to the underlying trajectories that are robust to this type of noise.
> > > > > - Second, the LAND metric — even if computed separately for each time subinterval — is inherently based on distance in space, without regard for the time variable. It is thus capable of capturing the local geometry (density in the vicinity of a time point) yet fails to capture the global geometry (evolution of marginals through time). However, our adversarial method is capable of capturing the complex global geometry, as is evident from our multiple experiments in the paper (e.g., Figures 2, 3, 7).
> > > > >
> > > > > Therefore, the main conclusion holds: **our method is a fundamentally novel approach to learning interpolants in multi-marginal flow matching which scales better with the growth of the number of marginal distributions**.
> > > > >
> > > > > Note that we have implemented the reviewer’s **second action point** by updating the second paragraph in the Related Works section, the OT-MFM trajectories in Fig. 2 and in §1 where OT-MFM now is described as a piecewise, time-dependent FM method.
> > > > >
> > > > > ### **In conclusion**
> > > > >
> > > > > The above results and discussion have been incorporated into the revised paper posted here on OpenReview, and we have taken care to ensure that the discussion accurately represents the baseline. As we have shown, even after accounting for the MFM implementation details noted by the reviewer, the main conclusions of the paper still hold.
> > > > >
> > > > > As such, we believe that our response addresses all the reviewer’s concerns. **Specifically, we have satisfied both of the *two actions* requested by the reviewer.**
> > > > >
> > > > > We would like to thank the reviewer once again for their valuable comments and the rigour with which they approached our paper.

---

### Official Review · Reviewer_1S64 · 2025-10-31

**Soundness:** 3
**Presentation:** 3
**Contribution:** 4
**Rating:** 8
**Confidence:** 4

**Summary:**

This paper tackles *multi-marginal trajectory inference* from unpaired snapshots observed at discrete times. The proposed method, ALI-CFM (Adversarially Learnt Interpolants + Conditional Flow Matching), departs from hand-crafted paths (linear, spline, piecewise-linear) by *learning* a time-conditioned interpolant $G_\phi$ whose intermediate-time pushforwards match the observed marginals.

**Strengths:**

* **(S1) Interpolant learning via distribution matching.** Adversarial learning of $G_\phi$ avoids brittle pointwise constraints and yields smoother, noise-robust paths.

* **(S2) Non-stationary geometry.** Clear wins on datasets with evolving topology/geometry (e.g., Knot, Cell Tracking), where time-independent metrics (e.g., MFM) underperform.

* **(S3) High $K$ stability.** Handles very large numbers of time points (e.g., $K=1200$) without the "kinks" typical of piecewise methods, easing CFM training.

* **(S4) Theory.** Uniqueness results for linear / piecewise-linear reference regularizers (Thms. 2.1–2.2) provide identifiability-style guarantees.

* **(S5) Spatial transcriptomics.** Strong results on a challenging multimodal ST task and competitive performance on cell tracking benchmarks.

**Weaknesses:**

* **(W1) Adversarial stability.** No analysis of GAN stability, failure modes (e.g., mode collapse), or sensitivity to $\lambda$, critic class, or choice of GAN loss.

* **(W2) Compute overhead.** Two-stage training plus the need to evaluate $\partial_t f_\phi$ during CFM adds cost; the paper lacks wall-clock and complexity breakdowns versus single-stage baselines.

* **(W3) High-dimensional scRNA-seq.** In 50D/100D, ALI-CFM is only on par with (or slightly below) OT-MFM; limited discussion of causes or remedies.

**Questions:**

1. **Stability of adversarial training (W1).** Did you observe mode collapse or time-wise imbalance across $\{t_i\}$? How sensitive are results to $\lambda$ and to the GAN loss (vanilla vs. R3GAN in Appx. D.3)? Any training heuristics that were essential (e.g., critic:generator update ratios, spectral norm, gradient penalties)?

2. **Compute and $\partial_t f_\phi$ (W2).**
   * Authors stated "little overhead" but would be great to report wall-clock and scaling with $(d,K,N)$ for ALI, CFM, and end-to-end ALI-CFM, compared to OT-MFM/OT-CFM.
   * Algorithm 2 requires $\partial_t f_\phi(x_0,x_1,t)$. Do you autograd through $t$ every CFM step? What is the measured overhead relative to settings with analytic $\dot{G}$?

3. **Regularizer selection.** You mix linear reference, piecewise-linear reference, and $\|\partial_{tt} G\|^2$ across datasets. Can you provide practical guidance (e.g., stationary vs. evolving geometry, large $K$, noise levels) and an ablation isolating each regularizer per dataset?

4. **High-dimensional performance (W3).** Beyond the pointwise-vs-distribution-matching explanation, what limits ALI-CFM in 50D/100D (critic capacity, coupling $\pi$, time-conditioning)? Have you tried stronger critics or multi-scale time features?

---

> ### Author Response · Authors · 2025-11-25
> **Addressing the stability of GAN training (Q1, W1)**
>
> We are happy that the reviewer appreciates our work and believes it should be accepted. Please find our response below.
>
> ### Addressing the stability of GAN training (Q1, W1)
>
> We would like to thank the reviewer for raising this important concern. We did not experience mode collapse in general; however, we found that it was crucial to use a multi-marginal (Markov-chained) OT plan in the ST experiment to get accurate interpolants. This is mentioned in our submission in the second-to-last paragraph in §4.4.
>
> There are indeed plenty of training heuristics that one could consider when training GANs; however, we did not find it necessary to extensively explore such techniques. E.g., we train the critic and generator net with equally many backpropagation passes.
>
> In relation to the choice of $\lambda$, we agree that carefully tuning this parameter is important. In the ablations below, we show how varying $\lambda$ affects the resulting EMD score, as well as the straightness of the trajectories. The straightness is measured using *path energy* defined in the following way:
>
> Path energy = $ E_{q_{0}}[\int_{0}^1\Vert u^\theta_{t}(x_t)\Vert^2 dt] $.
>
> The results of the ablation study are presented in the tables below:
>
> ### **Cite 50D Piecewise linear reference regulariser**
>
> **t = 1**
>
> | coefficient value | Interpolant (EMD) | CFM (EMD) | Path Energy |
> | --- | --- | --- | --- |
> | 0.01 | 46.96 | 47.29 | 300.01 |
> | 0.05 | 41.11 | 41.07 | 697.19 |
> | 0.1 | 40.76 | 41.19 | 1243.33 |
> | 0.2 | 41.21 | 42.20 | 2979.58 |
> | 0.5 | 41.82 | 40.93 | 5666.17 |
> | 5 | 42.23 | 42.59 | 36264.89 |
> | 10 | 41.38 | 43.81 | 87052.14 |
>
> **t = 2**
>
> | coefficient value | Interpolant (EMD) | CFM (EMD) | Path Energy |
> | --- | --- | --- | --- |
> | 0.01 | 38.12 | 42.17 | 358.98 |
> | 0.05 | 37.70 | 41.17 | 385.26 |
> | 0.1 | 37.28 | 40.51 | 551.10 |
> | 0.2 | 38.28 | 41.75 | 238.59 |
> | 0.5 | 37.00 | 39.48 | 268.40 |
> | 5 | 35.93 | 41.70 | 6297.56 |
> | 10 | 35.57 | 43.06 | 8484.26 |
>
> ### **Cite 50D Norm of second derivative regulariser**
>
> **t = 1**
>
> | coefficient value | Interpolant (EMD) | CFM (EMD) | Path Energy |
> | --- | --- | --- | --- |
> | 1.00E-06 | 41.50 | 42.17 | 12.30 |
> | 5.00E-07 | 41.48 | 42.14 | 12.00 |
> | 1.00E-07 | 41.44 | 42.06 | 12.56 |
> | 5.00E-08 | 41.38 | 42.23 | 13.91 |
> | 1.00E-08 | 40.75 | 41.59 | 26.00 |
> | 5.00E-09 | 41.39 | 42.24 | 41.98 |
> | 1.00E-09 | 42.89 | 43.81 | 56.39 |
>
> **t = 2**
>
> | coefficient value | Interpolant (EMD) | CFM (EMD) | Path Energy |
> | --- | --- | --- | --- |
> | 1.00E-06 | 39.63 | 45.66 | 7.58 |
> | 5.00E-07 | 39.55 | 45.51 | 7.55 |
> | 1.00E-07 | 39.35 | 45.23 | 7.68 |
> | 5.00E-08 | 39.31 | 44.99 | 8.10 |
> | 1.00E-08 | 39.06 | 41.61 | 13.21 |
> | 5.00E-09 | 39.04 | 40.14 | 18.51 |
> | 1.00E-09 | 41.24 | 40.04 | 39.65 |
>
> ### **Cite 100D Norm of second derivative regulariser**
>
> **t = 1**
>
> | coefficient value | Interpolant (EMD) | CFM (EMD) | Path Energy |
> | --- | --- | --- | --- |
> | 1.00E-06 | 47.00 | 49.89 | 6.90 |
> | 5.00E-07 | 47.03 | 49.37 | 6.66 |
> | 1.00E-07 | 47.03 | 49.65 | 10.02 |
> | 5.00E-08 | 46.87 | 49.04 | 11.25 |
> | 1.00E-08 | 46.66 | 48.36 | 25.43 |
> | 5.00E-09 | 46.52 | 47.84 | 30.93 |
> | 1.00E-09 | 46.97 | 47.92 | 55.48 |
>
> **t = 2**
>
> | coefficient value | Interpolant (EMD) | CFM (EMD) | Path Energy |
> | --- | --- | --- | --- |
> | 1.00E-06 | 44.78 | 51.12 | 6.90 |
> | 5.00E-07 | 44.88 | 50.92 | 6.60 |
> | 1.00E-07 | 44.80 | 50.99 | 7.09 |
> | 5.00E-08 | 44.77 | 51.19 | 11.86 |
> | 1.00E-08 | 44.65 | 49.98 | 31.18 |
> | 5.00E-09 | 44.74 | 50.00 | 37.22 |
> | 1.00E-09 | 45.97 | 49.60 | 28.22 |
>
> Interestingly, we see that increasing the regularising coefficient, combined with the piecewise linear regulariser, makes the trajectories less smooth. We argue that this is because the piecewise linearity forces interpolants to pass through marginal sample pointwise, akin to OT-CFM, MFMM and MFM with a time-dependent metric, and so the trajectories become less straight. Meanwhile, a stronger regularising weight combined with the other regularisers in general straightens the interpolants (decreases the path energy).

---

> ### Author Response · Authors · 2025-11-25
> **Addressing the compute overhead (W2, Q2)**
>
> We would like to note that the main baseline [1] also has a two-stage pipeline that necessitates computing $\partial_t f$. Since both ALI and MFM do require the time derivative of the interpolant, and modern autograd libraries (such as PyTorch) allow for efficient computation of autograd, we do not separately measure the time overhead. Training ALI-CFM for a single time step takes 40 minutes on our hardware, which is around the same time as it takes MFM to train.
>
> In addition, we would like to point out that our method is more advantageous if the number of intermediate marginal distributions is large. `Knot' distribution with 1000 marginals can be considered as an example. Then MFM requires RBF metric to be trained for every single marginal, while ALI-CFM is capable of dealing with the marginal distributions altogether without introducing any additional cost.
>
> We do not observe any serious overhead from computing the exact derivative $\partial_t f$ using autograd; therefore, we do not experiment with substituting it with a numerical estimation, as it would definitely trade the quality, yet produce little to know time benefits.
>
> References:
> [1] Kapusniak et al. (2024), *Metric flow matching for smooth interpolations on the data manifold*

---

> > ### Author Response · Authors · 2025-11-25
> > **Regulariser selection (Q3)**
> >
> > A general rule of thumb when selecting a regulariser is that in low-dimensional datasets with small numbers of modes in the marginal distributions, and when $K$ is large, the linear reference is a computationally efficient and generally sufficiently accurate reference. For instance, other regularisers did not notably improve our performances on the knot and cell tracking datasets.
> >
> > In high dimensions, on the other hand, the linear regulariser can be inaccurate, as stated in §2.3. This is probably due to the curse of dimensionality, which makes the mismatch between the support of the end-marginals $(q_0, q_1)$ and the intermediate marginals more severe.
> >
> > In low dimensions but with multi-modal marginal distributions, the linear reference can still work sufficiently well if it is combined with a multi-marginal OT (MMOT) plan. Using the MMOT plan relaxes the effect of multi-modality by coupling modes in the separate marginals that are close (according to the OT cost) throughout the pseudo-time axis.
> >
> > In the ST data, which is 2D but highly multi-modal, we used the piecewise linear reference interpolant to further ensure that the interpolants pass through meaningful regions of the tumour volume. This reduced the variance when training our ALIs.
> >
> > Thank you for posing these important questions that will guide future users of our method. We will make sure to include the explanations provided above in a dedicated appendix section of our submission.

---

> > > ### Author Response · Authors · 2025-11-25
> > > **High-dimensional performance (W3, Q4)**
> > >
> > > We believe that the discrepancy in performance between MFM / CFM and ALI can be attributed to the fact that the underlying geometry of the data in this benchmark is relatively simple and, therefore, can be successfully captured by the metric learning approach of MFM, whereas a GAN-based approach, although it successfully recovers the target distribution, evidently produces less accurate reconstruction, potentially due to the nature of adversarial training. We should also note that this comparison is conducted in the manner advantageous to the MFM, as the MFM learn a separate RBF metric for each time step, while we do not incorporate a learnable metric in our pipeline, which can also lead to a slightly degraded performance. While trying to achieve the best scores, we mostly focused on varying GAN loss and regularisation terms.

---

### Official Review · Reviewer_xv91 · 2025-11-03

**Soundness:** 3
**Presentation:** 3
**Contribution:** 2
**Rating:** 4
**Confidence:** 4

**Summary:**

This paper proposes a new approach to learn probability distributions dynamics with multiple marginals, based on flow matching. Recognizing that previous approaches to define the interpolants across multiple marginals (piecewise linear or splines) may exhibit high curvature, thereby leading to suboptimal training, the authors propose to *learn* the interpolants.

To learnt the interpolants, the authors enforce that the marginal distributions match at each time steps, using a adversarial loss in practice. Using the learnt interpolants, the authors then plug them directly into the flow matching machinery.

The authors propose multiple trajectories regularization for learning the interpolants, such as linearity, or curvature minimization.

The authors then show qualitatively that this results in smoother trajectories on synthetic and cell tracking data, leading to more accurate flow. Quantitatively, they evaluate their approach on cellular trajectories and tumor proliferation and demonstrate that their approach is competitive with state of the art methods.

**Strengths:**

- This paper proposes a creative approach to learning more meaningful interpolants in flow matching, which is of significant interest when dealing with multiple marginal distributions.
- I appreciate the design of the new experiment using spatial data, although the biological relevance of this experiment is questionable as it only takes into account the spatial distribution of the tumor, discarding all the single cell RNA seq resolution of the data.
- The paper is easy to read and the authors expose their ideas and contributions clearly.
- The authors demonstrate the performance of their method graphically, which makes the evaluation of performance of the method easier for the reader.

**Weaknesses:**

- An important motivation for this work is to have interpolant with less curvature, hopefully leading to improved training. As such, one experimental result that is lacking from this work is the cumulative curvature of the learnt interpolants, as well as some notion of divergence between the pushforward distribution with the GAN interpolants and the observed marginal distributions (for all different interpolant strategies). Ideally, depending on the hyperparameters and the strength of the regularization used, there would be some trade-off between the cumulative curvature and the matching at each time step. In my opinion, that would really nail down the contribution of the paper.

- The discriminator in the GAN seems to be common across all time steps. This seems suboptimal, especially for trajectories that intersect, like the synthetic knot experiment. In my opinion, you need a different discriminator for each time step, for the argument to hold, which may be challenging to train correctly when you don't have many samples per time step.

- The most established benchmark in this paper is the scRNAseq trajectory inference. However, this model does not improve on that benchmark. That undermines the main motivation for this paper. The spatial dataset is, from my understanding of biology, somewhat toy-ish too, as I expect there may much better suited methods for that problem that flow matching. If this model does not improve performance on tasks that are established to be meaningful, it raises the question of "what problem this paper is really solving?".

**Questions:**

- I’m surprised by how non-smooth the trajectories of the splines are in Figure 2. Did you compute splines between points that are matched with OT  between successive time steps ? Or abitrary points ? I think MMFM uses OT coupling between distributions.
- The authors state `Although our algorithm is on par with existing baselines, we believe that the nature of adversarial
training makes it difficult to completely outperform OT-MFM. Since our adversarially learned
interpolant matches the points at each time point in a distribution-matching sense, it might lose in
pointwise metrics to methods that are trained to overfit to the given points.` That is an interesting point, although the metric here is EMD so why do the authors refer to pointwise metrics ? Also, by the same argument as used by the authors, do we expect that this method may generate non-“realistic” samples ? That is, because the interpolants don't exactly align with the real data points, the model never learns to exactly generate "realistic" samples (it generates something slightly off). I would like the authors to comment on that as it can be a significant limitation of the method.
- Echoing my comment above: could you please comment on having a discriminator at each time step - whether you did this or not - and potential limitations in terms of sample size.
- Also echoing my other comment above, could the authors add the metrics I pointed to above - or  if not, could they argue why it's not relevant ? ` An important motivation for this work is to have interpolant with less curvature, hopefully leading to improved training. As such, one experimental result that is lacking from this work is the cumulative curvature of the learnt interpolants, as well as some notion of divergence between the pushforward distribution with the GAN interpolants and the observed marginal distributions (for all different interpolant strategies). Ideally, depending on the hyperparameters and the strength of the regularization used, there would be some trade-off between the cumulative curvature and the matching at each time step. In my opinion, that would really nail down the contribution of the paper.`
- I understand that this is very challenging give the time constraints, but I think it would be great to have an established benchmark where your method outperforms previous baselines. If not, please consolidate why the ST task is actually relevant biologically.

---

> ### Author Response · Authors · 2025-11-21
>
> We thank the reviewer for their insightful comments. We have addressed all the questions below, and we include a new ablation study of the different regularisers based on the reviewer’s request.
>
> ### **Request for cumulative curvature of the learnt interpolants experiment**
>
> We sincerely thank the reviewer for this request — we agree that adding these results will help *nail down* the important interplay of regularisers, regularising weights and the curvature of our trajectories. To this end, we have run extensive experiments to meet the reviewer’s request, and we share the results in the table below.
>
> ### **Cite 50D Piecewise linear reference regulariser**
>
> **t = 1**
>
> | coefficient value | Interpolant (EMD) | CFM (EMD) | Path Energy |
> | --- | --- | --- | --- |
> | 0.01 | 46.96 | 47.29 | 300.01 |
> | 0.05 | 41.11 | 41.07 | 697.19 |
> | 0.1 | 40.76 | 41.19 | 1243.33 |
> | 0.2 | 41.21 | 42.20 | 2979.58 |
> | 0.5 | 41.82 | 40.93 | 5666.17 |
> | 5 | 42.23 | 42.59 | 36264.89 |
> | 10 | 41.38 | 43.81 | 87052.14 |
>
> **t = 2**
>
> | coefficient value | Interpolant (EMD) | CFM (EMD) | Path Energy |
> | --- | --- | --- | --- |
> | 0.01 | 38.12 | 42.17 | 358.98 |
> | 0.05 | 37.70 | 41.17 | 385.26 |
> | 0.1 | 37.28 | 40.51 | 551.10 |
> | 0.2 | 38.28 | 41.75 | 238.59 |
> | 0.5 | 37.00 | 39.48 | 268.40 |
> | 5 | 35.93 | 41.70 | 6297.56 |
> | 10 | 35.57 | 43.06 | 8484.26 |
>
> ### **Cite 50D Norm of second derivative regulariser**
>
> **t = 1**
>
> | coefficient value | Interpolant (EMD) | CFM (EMD) | Path Energy |
> | --- | --- | --- | --- |
> | 1.00E-06 | 41.50 | 42.17 | 12.30 |
> | 5.00E-07 | 41.48 | 42.14 | 12.00 |
> | 1.00E-07 | 41.44 | 42.06 | 12.56 |
> | 5.00E-08 | 41.38 | 42.23 | 13.91 |
> | 1.00E-08 | 40.75 | 41.59 | 26.00 |
> | 5.00E-09 | 41.39 | 42.24 | 41.98 |
> | 1.00E-09 | 42.89 | 43.81 | 56.39 |
>
> **t = 2**
>
> | coefficient value | Interpolant (EMD) | CFM (EMD) | Path Energy |
> | --- | --- | --- | --- |
> | 1.00E-06 | 39.63 | 45.66 | 7.58 |
> | 5.00E-07 | 39.55 | 45.51 | 7.55 |
> | 1.00E-07 | 39.35 | 45.23 | 7.68 |
> | 5.00E-08 | 39.31 | 44.99 | 8.10 |
> | 1.00E-08 | 39.06 | 41.61 | 13.21 |
> | 5.00E-09 | 39.04 | 40.14 | 18.51 |
> | 1.00E-09 | 41.24 | 40.04 | 39.65 |
>
> ### **Cite 100D Norm of second derivative regulariser**
>
> **t = 1**
>
> | coefficient value | Interpolant (EMD) | CFM (EMD) | Path Energy |
> | --- | --- | --- | --- |
> | 1.00E-06 | 47.00 | 49.89 | 6.90 |
> | 5.00E-07 | 47.03 | 49.37 | 6.66 |
> | 1.00E-07 | 47.03 | 49.65 | 10.02 |
> | 5.00E-08 | 46.87 | 49.04 | 11.25 |
> | 1.00E-08 | 46.66 | 48.36 | 25.43 |
> | 5.00E-09 | 46.52 | 47.84 | 30.93 |
> | 1.00E-09 | 46.97 | 47.92 | 55.48 |
>
> **t = 2**
>
> | coefficient value | Interpolant (EMD) | CFM (EMD) | Path Energy |
> | --- | --- | --- | --- |
> | 1.00E-06 | 44.78 | 51.12 | 6.90 |
> | 5.00E-07 | 44.88 | 50.92 | 6.60 |
> | 1.00E-07 | 44.80 | 50.99 | 7.09 |
> | 5.00E-08 | 44.77 | 51.19 | 11.86 |
> | 1.00E-08 | 44.65 | 49.98 | 31.18 |
> | 5.00E-09 | 44.74 | 50.00 | 37.22 |
> | 1.00E-09 | 45.97 | 49.60 | 28.22 |
>
> In the tables, we have shared ablations on different dimensions of the CITE-seq, single-cell data, ablating regularisers, regulariser coefficients and the effect of these on EMDs and path energies. Path energy, here, quantifies the straightness of the trajectories (higher numbers, less straight paths). E.g., the path energy of the ALI-CFM trajectories is
>
> Path energy = $E_{q_0}[\int_{0}^1\Vert u^\theta_{t}(x_t)\Vert^2 dt]$.
>
> Interestingly, we see that increasing the regularising coefficient combined with the piecewise linear regulariser makes the trajectories less smooth. We argue that this is because the piecewise linearity forces interpolants to pass through marginal samples pointwise, akin to OT-CFM, MFMM and MFM with a time-dependent metric, and so the trajectories become less straight.
>
> Meanwhile, a stronger regularising weight combined with the other regularisers in general straightens the interpolants (decreases the path energy).
>
> ### **One discriminator per time step**
>
> The reviewer states: *The discriminator in the GAN seems to be common across all time steps. This seems suboptimal, especially for trajectories that intersect, like the synthetic knot experiment. In my opinion, you need a different discriminator for each time step, for the argument to hold, which may be challenging to train correctly when you don't have many samples per time step.*
>
> The reviewer also requests: *could you please comment on having a discriminator at each time step - whether you did this or not - and potential limitations in terms of sample size.*
>
> You are correct that there needs to be multiple discriminators, and not a single one — but for us, is achieved through amortisation, i.e., by **conditioning the discriminator on the considered time step**, allowing generalisation between nearby time steps. As such, our discriminators have shared weights and thus excellently handle low sample sizes (note that in the knot and cell tracking experiments, there are only 10 samples per marginal distribution/time step).

---

> > ### Author Response · Authors · 2025-11-21
> >
> > ### **3D ST experiment**
> >
> > **Regarding the biological relevance.** The reviewer asked us to *consolidate why the ST task is actually relevant biologically.*
> >
> > As mentioned in our submission, the ST dataset is obtained from the recent Nature publication *Tumour evolution and microenvironment interactions in 2D and 3D space,* by Mo et al. (Nature 2024). The design of our 3D tumour coordinate reconstruction task is directly motivated by Mo et al., who demonstrate that the spatial organisation of tumours (independently of single-cell transcriptional resolution) is itself a primary driver of clonal structure, immune infiltration, and therapeutic resistance.
> >
> > Their study shows that certain tumour fitness trademarks are inherently three-dimensional properties that cannot be deduced from scRNA-seq alone. Indeed, Mo et al. dedicate major analyses to **tumour purity**, boundaries, volumes, and 3D connectivity, underscoring that spatial density structure is a biologically meaningful signal. Note that the mentioned features here are all functions of scRNA-seq data aggregation, i.e., the scRNA-seq data has been marginalised out to obtain spatial distributions.
> >
> > Our experiment targets precisely this dimension of the biology: reconstructing the tumour’s 3D spatial density, annotated using the provided **tumour boundary scores**. This is a problem highlighted as highly relevant and technically challenging in Mo et al., and one that has broad translational value.
> >
> > **Regarding the statement “discarding the single cell RNA seq” data.** The reviewer states that the experiment *is questionable as it only takes into account the spatial distribution of the tumor, discarding all the single cell RNA seq resolution of the data*.
> >
> > Please note that the coordinates in the spatial distribution of tumour regions is annotated using the provided **tumour purity scores**, which are directly derived from the scRNA-seq data. So the scRNA-seq data is not discarded.
> >
> > **Regarding previous baselines.** Notably, Mo et al. devise a pipeline for 3D interpolation of tumour microenvironment boundaries and neighbourhoods using the tumor purity score (an scRNA-seq derived score) to annotate tumor regions. However, they do not directly interpolate the coordinates of the tumor annotated regions, and we did not find that their pipeline generalised to this task. As such, we could not use their pipeline as a baseline.
> >
> > Mo et al. use the PASTE software (Zeira et al., 2022) which is an OT method for aligning the ST slides. It minimizes a fused Gromov-Wasserstein objective on the joint space of expressions and intra-spot distances, and outputs a transport plan between spots in adjacent slides. However, this tool is used for aligning ST slides, and we could therefore have used it as a preprocessing tool. Instead, we aligned the slides based on the image data (see Appendix C for an in-depth explanation).
> >
> > After alignment, the transport map produced by PASTE would be equivalent to standard OT in Euclidean space. This is exactly the map that we are providing all benchmarked methods with in order to do OT-based conditional flow matching.
> >
> > In summary, we are unaware of other relevant methods that tackle this 3D ST problem.
> >
> > **Relevance as a flow matching application.** First, note that the idea of 3D modelling using multi-slide ST data is a new research field. As such, there are no clear alternatives to FM approaches available in the literature. In fact, the multi-marginal FM methodology fits the multi-slide ST perfectly, specifically in our experimental design: from each slide $i$ we observe samples from a marginal distribution $q_{t_i}$. Samples from $q_{t_i}$ with samples from any other slide $q_{t_j}$ (where $i\neq j$) are naturally unpaired, since the tissue shrinks or expands during storage, and the sequencing technology is destructive (the same cells can not be in multiple slides). Hence, the interpolations obtained from marginalisation of the learnt vector field are excellently posed to infer the dynamics of the tumour annotated regions.
> >
> > **Thank you for letting us clarify the biological and methodological relevance.** We are happy to use the extra page in the final version to further highlight the biological and methodological relevance of our experiment. We hope the reviewer now shares our enthusiasm for the new experimental design and its potential as a meaningful application of FM methods in high-impact spatial cancer research.

---

> > > ### Author Response · Authors · 2025-11-21
> > >
> > > ### **Questions**
> > >
> > > **Q1:** Thank you for bringing this up, we find the behaviour interesting but expected. We first clarify that, yes, the splines between points are matched with a multi-marginal OT plan with an additive cost assumption (following the MMFM paper), i.e., **not** arbitrary points/independent couplings.
> > >
> > > Now, it is expected that splines will produce non-smooth trajectories here since 1) there are few samples per marginal (only 10 per time step), 2) the time-adjacent marginal distributions are heavily overlapping in 2D space, and 3) the splines pass through marginal samples pointwise. As a result, the spline-based trajectories appear similarly “noisy” to the piecewise linear trajectories.
> > >
> > > **Q2:** We would like to use this important question to point out the elegant connection between classical GAN training and the adversarial interpolant learning that we propose:
> > >
> > > The typical data scenario motivating GAN training, is when one has access to a set of samples from some unknown distribution. Since GAN training is agnostic to the data-generating density/distribution, it neatly fits this scenario. Of relevance to the reviewer’s question, a trained GAN generates *realistic* samples — this is how the discriminator is fooled.
> > >
> > > In multi-marginal flow matching we are given samples from multiple marginal distributions, and, as in the GAN setting, we are agnostic to the density of these distributions. As such, adversarial learning is motivated in the MMFM problem setting and, based on the nature of GAN training, we expect to generate *realistic* interpolants as our ALIs are *approximately* distributed according to the underlying marginal distributions. Indeed, in, e.g., Fig. 8 we do observe our inferred trajectories to be by far the most realistic/accurate ones compared to the baselines.
> > >
> > > **Remaining questions:** Please see the answers above.
> > >
> > > **Let us know if there are any follow-up questions. We again thank the reviewer for the possibility to expand on the importance of our novel ST experiment, and for the recommended ablation study. If we have satisfactorily answered your questions, please consider updating your score.**

---

> ### Comment · Reviewer_xv91 · 2025-11-26
> **Thank you for your answer**
>
> I want to thank and commend the authors for addressing my comments and including the requested experiments. I have updated my score accordingly.

---

### Official Review · Reviewer_1Zg7 · 2025-11-07

**Soundness:** 3
**Presentation:** 3
**Contribution:** 2
**Rating:** 6
**Confidence:** 4

**Summary:**

This paper tackles multi-marginal flow matching from  intermediate time marginal data. The paper proposed a 2-stage approach, first learn a non-linear interpolant to match intermediate marginals via adversarial learning and then fit a vector field with the CFM loss on those interpolants. Experiments on synthetic, cell-tracking, scRNA-seq, and spatial transcriptomics are provided to demostrate the marginal matching property of the proposed method.

**Strengths:**

1. The paper is clearly written and easy to understand.
2. Theoretical analysis showing existence of the interpolants strengthens the paper.
3. Practical benefits are demonstrated in the experiments.

**Weaknesses:**

1. Compared to MFM, the method differs in use of GAN based distribution matching for intermediate marginals. The added complexity of adversarial training makes the idea less appealing compared to existing methods that do not require adversarial training.
2. The performance on Section 4.3 single-cell RNA-seq seems to be weak compared to MFM that does not require adversarial training.
3. Only OT coupling based implementation is used in the experiments. OT adds additional complexity and is challenging to compute in high dimension.
4. Ablations are not provided for OT/Non-OT, different regularizations, etc.
5. The empirical validation is limited to low dimensional applications.

**Questions:**

1. Could the authors explain the reason behind weaker performance in Section 4.3 ?
2. Can the method work well with independent coupling ? How does it compare to OT-based? What would be the challenges?
3. Could the authors provide ablation study for different regularizations?
4. Could you discuss the possible challenges for high dimensional applications ?

---

> ### Author Response · Authors · 2025-11-21
>
> We thank the reviewer for their careful comments. Below, we have addressed all the questions, and we include a new ablation study of the different regularisers.
>
> ### **Regarding complexity of adversarial training**
>
> We wish to point out that adversarial training does not necessarily add complexity. In fact, to showcase this in relation to MFM, we attempted to run learn time-dependent RBF metrics on our knot experiment — recall that there are more than 1000 marginal distributions here. Solving this problem with MFM and time-dependent RBF metrics amounts to training more than 1000 RBF networks, each trained using gradient descent. Merely fitting the metrics took around **6.5 hrs**. Notably, this is a 2D experiment with 10 samples per marginal distribution. In contrast, our ALIs were fitted in ca. 40 minutes without early stopping, and with no mentionable hyperparameter tuning.
>
> Clearly, adversarial training in this setting **reduces complexity** compared to the available baselines.
>
> ### **Regarding data dimensions**
>
> We respectfully disagree with the claim that the method is only evaluated in low dimensions as the single cell experiments are in 50D and 100D. In fact, the flavour of our experiments is highly aligned with the standard practice in the multi-marginal flow matching field. For instance the MMFM paper (Rohbeck et al., 2025) experimented with maximum 25 dimensions (see §5.2 in that paper). There is indeed an image translation experiment in the MFM paper (Kapusniak et al., 2024), but the MFM algorithm is employed in a much lower-dimensional VAE latent space. Other relevant articles where the main multi-marginal experiment is performed on the same single-cell data as we use include Tong et al. (2024), Neklyudov et al. (2024), Lee et al. (2025), Park and Lee (2025).
>
> **Mentioned references**
>
> Rohbeck et al. (2025), *Modeling complex system dynamics with flow matching
> across time and conditions*
>
> Kapusniak et al. (2024), *Metric flow matching for smooth interpolations
> on the data manifold*
>
> Tong et al. (2024), *Improving and generalizing flow-based generative models with minibatch optimal transport*
>
> Neklyudov et al. (2024), *A computational framework for solving Wasserstein lagrangian flows*
>
> Lee et al. (2025), *Multi-marginal stochastic flow matching for high-dimensional snapshot data at irregular time points*
>
> Park and Lee (2025), *Multi-marginal schrödinger bridge matching*
>
> ### **Questions**
>
> **Q1:** We believe that this can be attributed to the fact that the underlying geometry of the data in this benchmark is relatively simple and, therefore, can be successfully captured by the metric learning approach of MFM, whereas a GAN-based approach, although it successfully recovers the target distribution, evidently produces less accurate reconstruction, potentially due to the nature of adversarial training. We should also note that this comparison is conducted in the manner advantageous to the MFM, as the MFM learn a separate RBF metric for each time step, while we do not incorporate a learnable metric in our pipeline, which can also lead to a slightly degraded performance.
>
> However, we successfully show in the paper that MFM poorly scales to the setting when the geometry contains loops or when the number of intermediate time marginals in large. In contrast, our method works stably and efficiently in this setting.

---

> > ### Author Response · Authors · 2025-11-21
> >
> > **Q2:** This is an excellent question that allows us to highlight one of the multiple benefits of our method over existing multi-marginal flow matching methods. Consider an experimental setup with *many* marginal distributions, i.e. akin to our cell tracking or knot experiments where $K$ is large. If the number of samples from each of the marginals is sufficiently large such that the multi-marginal transport map cannot be pre-computed, then existing methods will need to compute the multi-marginal OT map in every training iteration. This will of course be prohibitively costly. ALI-CFM, on the other hand, can be trained merely by pairs of end-marginal samples. That is, ALI-CFM relies on a coupling over $(x_0, x_1)$, while the piecewise baseline methods require a coupling over $(x_{t_1}, x_{t_2},..., x_{t_K})$ where $t_1 = 0$ and $t_K=1.$
> >
> > The reviewer asked how ALI-CFM compares when using an independent and vs. an OT coupling. We have submitted a revised submission pdf where we in Figure 7 show the effects of using an independent coupling.  As expected, the CFM losses show the increased variance in the CFM objective when using the independent coupling compared to the OT coupling. The increased variance makes the I-ALI trajectories overlap to a greater extent than the OT-ALI trajectories.
> >
> > Regarding challenges of using an independent coupling in our work, in the spatial transcriptomics experiment, an independent coupling would be highly inefficient due to the multi-modal nature of the data. That is, if samples from adjacent slides would be independently/randomly coupled, all methods (N.B. this is not an ALI-related issue) would consider trajectories with end points from distinct modes, leading to high objective variance. For instance, I-CFM would regress to interpolants that, most of the times, would be far outside the tumor annotated regions in the data. As such, we did not consider the independent coupling in this experiment. We thank the reviewer for this inquiry, and will update the main text with a motivation based on what we explained above.
> >
> > **Q3:** This was an excellent idea, for which we thank the reviewer. In the tables below, we have conducted ablations on multiple single-cell data settings, ablating regularisers, regulariser coefficients and the effect of these on EMDs and path energies. Path energy, here, quantifies the straightness of the trajectories (higher numbers, less straight paths). Interestingly, we see that increasing the regularising coefficient combined with the piecewise linear regulariser makes the trajectories less smooth. We argue that this is because the piecewise linearity forces interpolants to pass through marginal samples pointwise, akin to OT-CFM, MFMM, and MFM with a time-dependent metric, and so the trajectories become less straight.
> >
> > ### **Cite 50D Piecewise linear reference regulariser**
> > **t = 1**
> > | coefficient value | Interpolant (EMD) | CFM (EMD) | Path Energy |
> > | --- | --- | --- | --- |
> > | 0.01 | 46.96 | 47.29 | 300.01 |
> > | 0.05 | 41.11 | 41.07 | 697.19 |
> > | 0.1 | 40.76 | 41.19 | 1243.33 |
> > | 0.2 | 41.21 | 42.20 | 2979.58 |
> > | 0.5 | 41.82 | 40.93 | 5666.17 |
> > | 5 | 42.23 | 42.59 | 36264.89 |
> > | 10 | 41.38 | 43.81 | 87052.14 |
> >
> > **t = 2**
> > | coefficient value | Interpolant (EMD) | CFM (EMD) | Path Energy |
> > | --- | --- | --- | --- |
> > | 0.01 | 38.12 | 42.17 | 358.98 |
> > | 0.05 | 37.70 | 41.17 | 385.26 |
> > | 0.1 | 37.28 | 40.51 | 551.10 |
> > | 0.2 | 38.28 | 41.75 | 238.59 |
> > | 0.5 | 37.00 | 39.48 | 268.40 |
> > | 5 | 35.93 | 41.70 | 6297.56 |
> > | 10 | 35.57 | 43.06 | 8484.26 |
> >
> > ### **Cite 50D Norm of second derivative regulariser**
> > **t = 1**
> > | coefficient value | Interpolant (EMD) | CFM (EMD) | Path Energy |
> > | --- | --- | --- | --- |
> > | 1.00E-06 | 41.50 | 42.17 | 12.30 |
> > | 5.00E-07 | 41.48 | 42.14 | 12.00 |
> > | 1.00E-07 | 41.44 | 42.06 | 12.56 |
> > | 5.00E-08 | 41.38 | 42.23 | 13.91 |
> > | 1.00E-08 | 40.75 | 41.59 | 26.00 |
> > | 5.00E-09 | 41.39 | 42.24 | 41.98 |
> > | 1.00E-09 | 42.89 | 43.81 | 56.39 |
> >
> > **t = 2**
> > | coefficient value | Interpolant (EMD) | CFM (EMD) | Path Energy |
> > | --- | --- | --- | --- |
> > | 1.00E-06 | 39.63 | 45.66 | 7.58 |
> > | 5.00E-07 | 39.55 | 45.51 | 7.55 |
> > | 1.00E-07 | 39.35 | 45.23 | 7.68 |
> > | 5.00E-08 | 39.31 | 44.99 | 8.10 |
> > | 1.00E-08 | 39.06 | 41.61 | 13.21 |
> > | 5.00E-09 | 39.04 | 40.14 | 18.51 |
> > | 1.00E-09 | 41.24 | 40.04 | 39.65 |

---

> > > ### Author Response · Authors · 2025-11-21
> > >
> > > ### **Cite 100D Norm of second derivative regulariser**
> > > **t = 1**
> > >
> > > | coefficient value | Interpolant (EMD) | CFM (EMD) | Path Energy |
> > > | --- | --- | --- | --- |
> > > | 1.00E-06 | 47.00 | 49.89 | 6.90 |
> > > | 5.00E-07 | 47.03 | 49.37 | 6.66 |
> > > | 1.00E-07 | 47.03 | 49.65 | 10.02 |
> > > | 5.00E-08 | 46.87 | 49.04 | 11.25 |
> > > | 1.00E-08 | 46.66 | 48.36 | 25.43 |
> > > | 5.00E-09 | 46.52 | 47.84 | 30.93 |
> > > | 1.00E-09 | 46.97 | 47.92 | 55.48 |
> > >
> > > **t = 2**
> > >
> > > | coefficient value | Interpolant (EMD) | CFM (EMD) | Path Energy |
> > > | --- | --- | --- | --- |
> > > | 1.00E-06 | 44.78 | 51.12 | 6.90 |
> > > | 5.00E-07 | 44.88 | 50.92 | 6.60 |
> > > | 1.00E-07 | 44.80 | 50.99 | 7.09 |
> > > | 5.00E-08 | 44.77 | 51.19 | 11.86 |
> > > | 1.00E-08 | 44.65 | 49.98 | 31.18 |
> > > | 5.00E-09 | 44.74 | 50.00 | 37.22 |
> > > | 1.00E-09 | 45.97 | 49.60 | 28.22 |
> > >
> > > **Q4:** We thank the reviewer for the opportunity to expand on this interesting topic. The main thing to consider when applying ALI in higher dimensions is the choice of regulariser. As we write above Equation (11) in our submission, when *the supports of the intermediate marginal
> > > distributions differ from those of the end marginals, the linear reference […] may restrict $G_\phi$ from accurate modelling of the target distributions, especially in high dimensions.* In exceptionally high dimensions we would furthermore recommend consulting the rich GAN optimisation literature for training techniques (we list such references in §5).
> > >
> > > **Let us know if there are any follow-up questions, and if we have satisfactorily answered, please consider updating your score.**

---

> > > > ### Author Response · Authors · 2025-11-27
> > > > **Follow-up response**
> > > >
> > > > We want to thank the reviewer once again for their review, and valuable time.
> > > >
> > > > **The deadline for the discussion phase is approaching. As such, we are wondering if the reviewer’s comments have been satisfactorily answered? If so, we kindly ask the reviewer to consider raising their score.**
> > > >
> > > > To ease the exposition of what we have achieved during the discussion period, we have now submitted a revision of our submission here on OpenReview. While respecting the 10-page limit, we have specifically integrated the following updates into our revised submission:
> > > >
> > > > - In §D.2 we compare the effects of independent and OT couplings, reporting and analysing the performances of I-ALI-CFM and OT-ALI-CFM on the cell tracking data.
> > > > - In §4.3 and §D.3.1 we provide an ablation study on different dimensions of the CITE-seq data, ablating our regularisers, regulariser coefficients and the effect of these on EMDs and trajectory straightness.
> > > > - In Discussion (§5) we provide a discuss regarding the complexity of adversarial training.
> > > > - Above Equation (5) we clearly accredit previous works for the considered interpolant parametrisation.
> > > >
> > > > Once again, we thank the reviewer for their valuable time.

---

### Meta-Review · Area_Chair_WYku · 2025-12-25

**Summary:**

This paper proposed a GAN-inspired loss to learn source to target interpolants matching the observed data points, and after using them to train a flow matching algorithm.
This paper got overall borderline scores where the main pros and cons raised in the reviews are summarized below:
+ Research focus on the interpolants is natural and potentially useful.
+ Well written paper.
+ Theoretical analysis of uniqueness of interpolants.
+ Benefits of interpolants in experiments.
- Added complexity in adversarial and OT training, 2 stage training, missing timing/complexity analysis.
- Potentially limited novelty (MFM).
- Missing Ablations, evaluations, e.g., properties of learned paths, GAN time sharing.
- scRNAseq experiments designed for the problem tackled in paper are a bit weak and/or with questionable biological significance.

One of the negative reviewers (out of total of 2) mentioned they updated their score but did not mention what target score. Their concern was around the biological significant, and missing evaluations. Overall it seems the paper is above the acceptance bar for ICLR.

**Reviewer Concerns:**

Please see above.

**Reviewer Scores:**

please see above.

---

### Decision · Program_Chairs · 2026-01-26

Accept (Poster)